# Universal Exact Compression of Differentially Private Mechanisms

**Yanxiao Liu**
The Chinese University of Hong Kong
yanxiaoliu@link.cuhk.edu.hk

**Wei-Ning Chen**
Stanford University
wnchen@stanford.edu

**Ayfer Özgür**
Stanford University
aozgur@stanford.edu

**Cheuk Ting Li**
The Chinese University of Hong Kong
ctli@ie.cuhk.edu.hk

## Abstract

To reduce the communication cost of differential privacy mechanisms, we introduce a novel construction, called Poisson private representation (PPR), designed to compress and simulate any local randomizer while ensuring local differential privacy. Unlike previous simulation-based local differential privacy mechanisms, PPR exactly preserves the joint distribution of the data and the output of the original local randomizer. Hence, the PPR-compressed privacy mechanism retains all desirable statistical properties of the original privacy mechanism such as unbiasedness and Gaussianity. Moreover, PPR achieves a compression size within a logarithmic gap from the theoretical lower bound. Using the PPR, we give a new order-wise trade-off between communication, accuracy, central and local differential privacy for distributed mean estimation. Experiment results on distributed mean estimation show that PPR consistently gives a better trade-off between communication, accuracy and central differential privacy compared to the coordinate subsampled Gaussian mechanism, while also providing local differential privacy.

## 1 Introduction

In modern data science, there is a growing dependence on large amounts of high-quality data, often generated by edge devices (e.g., photos and videos captured by smartphones, or messages hosted by social networks). However, this data inherently contains personal information, making it susceptible to privacy breaches during acquisition, collection, or utilization. For instance, despite the significant recent advancement in foundational models [9], studies have shown that these models can accidentally memorize their training data. This poses a risk where malicious users, even with just API access, can extract substantial portions of sensitive information [14, 15]. In recent years, differential privacy (DP) [28] has emerged as a powerful framework for safeguarding users' privacy by ensuring that local data is properly randomized before leaving users' devices. Apart from privacy concerns, communicating local data from edge devices to the central server often becomes a bottleneck in the system pipeline, especially with high-dimensional data common in many machine learning scenarios. This leads to the following fundamental question: how can we efficiently communicate privatized data?

Recent works have shown that a wide range of differential privacy mechanisms can be "simulated" and "compressed" using shared randomness, resulting in a "compressed mechanism" which has a smaller communication cost compared to the original mechanism, while retaining the (perhaps slightly weakened) privacy guarantee. This can be done via rejection sampling [31], importance sampling [71, 78], or dithered quantization [56, 72, 46, 49, 80] with each approach having its own advantages and disadvantages. For example, importance-sampling-based methods [71, 78] and the

38th Conference on Neural Information Processing Systems (NeurIPS 2024).

rejection-sampling-based method [31] can simulate a wide range of privacy mechanisms; however, the output distribution of the induced mechanism does not perfectly match the original mechanism. This is limiting in scenarios where the original mechanism is designed to satisfy some desired statistical properties, e.g. it is often desirable for the local randomizer to be unbiased or to be "summable" noise such as Gaussian or other infinitely divisible distributions. Since the induced mechanism is different from the original one, these statistical properties are not preserved. On the other hand, dithered-quantization-based approaches [48, 56, 72, 46, 49, 80] can ensure a correct simulated distribution, but they can only simulate additive noise mechanisms. More importantly, dithered quantization relies on shared randomness between the user and the server, and the server needs to know the dither for decoding. This annuls the local privacy guarantee on the user data, unless we are willing to assume a trusted aggregator [46], use an additional secure aggregation step [49], or restrict attention to specific privacy mechanisms (e.g., one-dimensional Laplace [72]).

**Our contribution**

In this paper, we introduce a novel "DP mechanism compressor" called *Poisson private representation (PPR)*, designed to compress and *exactly* simulate *any* local randomizer while ensuring local DP, through the use of shared randomness.[1] We elaborate on three main advantages of PPR, namely universality, exactness and communication efficiency.

**Universality.** Unlike dithered-quantization-based approaches which can only simulate additive noise mechanisms, PPR can simulate any local or central DP mechanism with discrete or continuous input and output. Moreover, PPR is *universal* in the sense that the user and the server only need to agree on the output space and a proposal distribution, and the user can simulate any DP mechanism with the same output space. The user can choose a suitable DP mechanism and privacy budget according to their communication bandwidth and privacy requirement, without divulging their choice to the server.

**Exactness.** Unlike previous DP mechanism compressors such as Feldman and Talwar [31], Shah et al. [71], Triastcyn et al. [78], PPR enables *exact* simulation, ensuring that the reproduced distribution perfectly matches the original one. Exact distribution recovery offers several advantages. Firstly, the compressed sample maintains the same statistical properties as the uncompressed one. If the local randomizer is unbiased (a crucial requirement for many machine learning tasks like DP-SGD), the outcome of PPR remains unbiased. In contrast, reconstruction distributions in prior simulation-based compression methods [31, 71] are often biased unless specific debiasing steps are performed (only possible for certain DP mechanisms [71]). Secondly, when the goal is to compute the mean (e.g., for private mean or frequency estimation problems) and the local noise is "summable" (e.g., Gaussian noise or other infinitely divisible distributions [55, 42]), exact distribution recovery of the local noise enables precise privacy accounting for the final *central* DP guarantee, without relying on generic privacy amplification techniques like shuffling [30, 32]. PPR can compress a central DP mechanism (e.g., the Gaussian mechanism [27]) and simultaneously achieve weaker local DP (i.e., with a larger $\varepsilon_{\mathsf{local}}$) and stronger central DP (i.e., with a smaller $\varepsilon_{\mathsf{central}}$), while maintaining exactly the same privacy-utility trade-offs as the uncompressed Gaussian mechanism.

**Communication efficiency.** PPR compresses the output of any DP mechanism to a size close to the theoretical lower bound. For a mechanism on the data $X$ with output $Z$, the compression size of PPR is $I(X; Z) + \log(I(X; Z) + 1) + O(1)$, with only a logarithmic gap from the mutual information lower bound $I(X; Z)$.[2] The "$O(1)$" constant can be given explicitly in terms of a tunable parameter $\alpha > 1$ which controls the trade-off between compression size, computational time and privacy.

The main technical tool we utilize for PPR is the Poisson functional representation [61, 60], which provides precise control over the reconstructed joint distribution in channel simulation problems [6, 45, 61, 35, 41, 10, 7, 21]. Channel simulation aims to achieve the minimum communication for simulating a channel (i.e., a specific conditional distribution). Typically, these methods rely on shared randomness between the user and server, and privacy is only preserved *when the shared randomness is hidden from the adversary*. This setup conflicts with local DP, where the server (which requires access to shared randomness for decoding) is considered adversarial. To ensure local DP, we introduce a randomized encoder based on the Poisson functional representation, which stochastically maps

---

[1]Our code can be found in `https://github.com/cheuktingli/PoissonPrivateRepr`

[2]This is similar to channel simulation [45] and the strong functional representation lemma [61], though [45, 61] do not concern privacy.

a private local message to its representation. Hence, PPR achieves order-wise trade-offs between privacy, communication, and accuracy, while preserving the original distribution of local randomizers.

**Notations.** Entropy $H(X)$, mutual information $I(X;Y)$, KL divergence $D(P\|Q)$ and logarithm are to the same base, e.g., they can be all in bits (base 2), or all in nats (base $e$). For $P, Q$, $\mathrm{d}P(\cdot)/\mathrm{d}Q$ denotes the Radon-Nikodym derivative.

## 2 Related Work

**Generic compression of local DP mechanisms.** In this work, we consider both central DP [28] and local DP [79, 53]. Recent research has explored methods for compressing local DP randomizers when shared randomness is involved. For instance, when $\varepsilon \leq 1$, Bassily and Smith [5] demonstrated that a single bit can simulate any local DP randomizer with a small degradation of utility, as long as the output can be computed using only a subset of the users' data. Bun et al. [12] proposed another generic compression technique based on rejection sampling, which compresses a $\varepsilon$-DP mechanism into a $10\varepsilon$-DP mechanism. Feldman and Talwar [31] proposed a distributed simulation approach using rejection sampling with shared randomness, while Shah et al. [71], Triastcyn et al. [78] utilized importance sampling (or more specifically, minimum random coding [20, 74, 47]). However, all these methods only *approximate* the original local DP mechanism, unlike our scheme, which achieves an *exact* distribution recovery.

**Distributed mean estimation under DP.** Mean estimation is the canonical problems in distributed learning and analytics. They have been widely studied under privacy [24, 8, 23, 4], communication [40, 11, 75], or both constraints [18, 31, 71, 43, 17, 19]. Among them, Asi et al. [4] has demonstrated that the optimal unbiased mean estimation scheme under local differential privacy is privUnit [8]. Subsequently, communication-efficient mechanisms introduced by Feldman and Talwar [31], Shah et al. [71], Isik et al. [51] aimed to construct communication-efficient versions of privUnit, either through distributed simulation or discretization. However, these approaches only approximate the privUnit distribution, while our proposed method ensures exact distribution recovery.

**Distributed channel simulation.** Our approach relies on the notion of channel simulation [6, 45, 61, 35, 41, 10, 7, 21]. One-shot channel simulation is a lossy compression task, which aims to find the minimum amount of communications over a noiseless channel that is in need to "simulate" some channel $P_{Z|X}$ (a specific conditional distribution). By Harsha et al. [45], Li and El Gamal [61], the average communication cost is $I(X; Z) + O(\log(I(X; Z)))$. In [45], algorithms based on rejection sampling are proposed, and it is further generalized in [39] by introducing the greedy rejection coding. Dithered quantization [81] has also been used to simulate an additive noise channel in [2] for neural compression. As also shown in [2], the time complexity of channel simulation protocols (e.g., in [61]) is usually high, and [76, 35, 41] try to improve the runtime under certain assumptions. Moreover, channel simulation tools have also been used in neural network compression [47], image compression via variational autoencoders [37], diffusion models with perfect realism [77] and differentially private federated learning [71].

**Poisson functional representation.** The Poisson functional representation is a channel simulation scheme studied in [61]. Also refer to [65] for related constructions for Monte Carlo simulations. Based on the Poisson functional representation, the Poisson matching lemma has been used in proving one-shot achievability results for various network information theory problems [60, 64]. Also see applications on unequal message protection [54], hypothesis testing [44], information hiding [63], minimax learning [62] and secret key generation [50]. A variation called the importance matching lemma [69] has also used in distributed lossy compression. By [38], the Poisson functional representation can be viewed as a certain variant of the $A^*$ sampling [66, 65], and hence an optimized version with better runtime for one-dimensional unimodal distribution has been proposed in [38].

## 3 Preliminaries

We begin by reviewing the formal definitions of differential privacy (DP). We consider two models of DP data analysis. In the central model, introduced in Dwork et al. [28], the data of the individuals is stored in a database $X \in \mathcal{X}$ by the server. The server is then trusted to perform data analysis

whose output $Z = \mathcal{A}(X) \in \mathcal{Z}$ (where $\mathcal{A}$ is a randomized algorithm), which is sent to an untrusted data analyst, does not reveal too much information about any particular individual's data. While this model requires a higher level of trust than the local model, it is possible to design significantly more accurate algorithms. We say that two databases $X, X' \in \mathcal{X}$ are neighboring if they differ in a single data point. More generally, we can consider a symmetric neighbor relation $\mathcal{N} \subseteq \mathcal{X}^2$, and regard $X, X'$ as neighbors if $(X, X') \in \mathcal{N}$.

On the other hand, in the local model, each individual (or client) randomizes their data before sending it to the server, meaning that individuals are not required to trust the server. A local DP mechanism [53] is a local randomizer $\mathcal{A}$ that maps the local data $X \in \mathcal{X}$ to the output $Z = \mathcal{A}(X) \in \mathcal{Z}$. Note that here $X$ is the data at one user, unlike central-DP where $X$ is the database with the data of all users. We now review the notion of $(\varepsilon, \delta)$-central and local DP.

**Definition 3.1** (Differential privacy [28, 53]). Given a mechanism $\mathcal{A}$ which induces the conditional distribution $P_{Z|X}$ of $Z = \mathcal{A}(X)$, we say that it satisfies $(\varepsilon, \delta)$-DP if for any neighboring $(x, x') \in \mathcal{N}$ and $\mathcal{S} \subseteq \mathcal{Z}$, it holds that

$$\Pr(Z \in \mathcal{S} \mid X = x) \leq e^{\varepsilon} \Pr(Z \in \mathcal{S} \mid X = x') + \delta.$$

In particular, if $\mathcal{N} = \mathcal{X}^2$, we say that the mechanism satisfies $(\varepsilon, \delta)$-local DP [53].[3]

When a mechanism satisfies $(\varepsilon, 0)$-central/local DP, we will refer to it simply as $\varepsilon$-central/local DP. $\varepsilon$-DP can be generalized to *metric privacy* by considering a metric $d_{\mathcal{X}}(x, x')$ over $\mathcal{X}$ [16, 3].

**Definition 3.2** ($\varepsilon \cdot d_{\mathcal{X}}$-privacy [16, 3]). Given a mechanism $\mathcal{A}$ with conditional distribution $P_{Z|X}$, and a metric $d_{\mathcal{X}}$ over $\mathcal{X}$, we say that $\mathcal{A}$ satisfies $\varepsilon \cdot d_{\mathcal{X}}$-privacy if for any $x, x' \in \mathcal{X}, \mathcal{S} \subseteq \mathcal{Z}$, we have

$$\Pr(Z \in \mathcal{S} \mid X = x) \leq e^{\varepsilon \cdot d_{\mathcal{X}}(x,x')} \Pr(Z \in \mathcal{S} \mid X = x').$$

This recovers the original $\varepsilon$-central DP by considering $d_{\mathcal{X}}$ to be the Hamming distance among databases, and recovers the original $\varepsilon$-local DP by considering $d_{\mathcal{X}}$ to be the discrete metric [16].

The reason we use $X$ to refer to both the database in central DP and the user's data in local DP is that our proposed method can compress both central and local DP mechanisms in exactly the same manner. In the following sections, the mechanism $\mathcal{A}$ to be compressed (often written as a conditional distribution $P_{Z|X}$) can be either a central or local DP mechanism, and the neighbor relation $\mathcal{N}$ can be any symmetric relation. The "encoder" refers to the server in central DP, or the user in local DP. The "decoder" refers to the data analyst in central DP, or the server in local DP.

## 4 Poisson Private Representation

**Definition 4.1** (Poisson functional representation [61, 60]). Let $(T_i)_i$ be a Poisson process with rate $1$ (i.e., $T_1, T_2 - T_1, T_3 - T_2, \ldots \overset{iid}{\sim} \mathrm{Exp}(1)$), independent of $Z_i \overset{iid}{\sim} Q$ for $i = 1, 2, \ldots$. Then $(Z_i, T_i)_i$ is a Poisson process with intensity measure $Q \times \lambda_{[0,\infty)}$ [57], where $\lambda_{[0,\infty)}$ is the Lebesgue measure over $[0, \infty)$. Fix any distribution $P$ over $\mathcal{Z}$ that is absolutely continuous with respect to $Q$. Let

$$\tilde{T}_i := T_i \cdot \left( \frac{\mathrm{d}P}{\mathrm{d}Q}(Z_i) \right)^{-1}. \tag{1}$$

Then $(Z_i, \tilde{T}_i)$ is a Poisson process with intensity measure $P \times \lambda_{[0,\infty)}$, which is from the mapping theorem [57]. The *Poisson functional representation* (PFR) [61, 60] selects the point $Z = Z_K$ with the smallest associated $\tilde{T}_K$, i.e., let $K := \arg\min_i \tilde{T}_i$ and $Z := Z_K$.[4]

The PFR selects a sample following the target distribution $P$ using another distribution $Q$. It draws a random sequence $(Z_i)_i$ from $Q$ and a sequence of times $(T_i)_i$ according to a Poisson process. If we select the sample $Z_i$ with the smallest $T_i$, then the selected sample follows $Q$. To obtain a sample from $P$ instead, we multiply the time by the factor $(\frac{\mathrm{d}P}{\mathrm{d}Q}(Z_i))^{-1}$ in (1) to give $\tilde{T}_i$, so the $Z_i$ with the smallest $\tilde{T}_i$ will follow $P$.

---

[3]Equivalently, local DP can be viewed as a special case of central DP with dataset size $n = 1$.

[4]Since the $T_i$'s are continuous, with probability $1$, there do not exist two equal values among $\tilde{T}_i$'s.

The Poisson functional representation guarantees that $Z \sim P$ [61]. To simulate a DP mechanism with a conditional distribution $P_{Z|X}$ using the Poisson functional representation, we can use $(Z_i)_i$ as the shared randomness between the encoder and the decoder. [5] Upon observing $X$, the encoder generates the Poisson process $(T_i)_i$, computes $\tilde{T}_i$ and $K$ using $P = P_{Z|X}$, and transmits $K$ to the decoder. The decoder simply outputs $Z_K$, which follows the conditional distribution $P_{Z|X}$. The issue is that $K$ is a function of $X$ and the shared randomness $(Z_i, T_i)_i$, and a change of $X$ may affect $K$ in a deterministic manner, and hence this method cannot be directly used to protect the privacy of $X$.

**Poisson private representation.** To ensure privacy, we introduce randomness in the encoder by a generalization of the Poisson functional representation, which we call *Poisson private representation (PPR)* with parameter $\alpha \in (1, \infty]$, proposal distribution $Q$ and the simulated mechanism $P_{Z|X}$. Both $X$ and $Z$ can be discrete or continuous, though as a regularity condition, we require $P_{Z|X}(\cdot|X)$ to be absolutely continuous with respect to $Q$ almost surely. The PPR-compressed mechanism is given as:

1. We use $(Z_i)_{i=1,2,\ldots}$, $Z_i \overset{iid}{\sim} Q$ as the shared randomness between the encoder and the decoder.

   Practically, the encoder and the decoder can share a random seed and generate $Z_i \overset{iid}{\sim} Q$ from it using a pseudorandom number generator.[6]

2. The encoder knows $(Z_i)_i, X, P_{Z|X}$ and performs the following steps:

   (a) Generates the Poisson process $(T_i)_i$ with rate 1.

   (b) Computes $\tilde{T}_i := T_i \cdot (\frac{dP}{dQ}(Z_i))^{-1}$, where $P := P_{Z|X}(\cdot|X)$. Take $\tilde{T}_i = \infty$ if $\frac{dP}{dQ}(Z_i) = 0$.

   (c) Generates $K \in \mathbb{Z}_+$ using local randomness with

   $$\Pr(K = k) = \frac{\tilde{T}_k^{-\alpha}}{\sum_{i=1}^{\infty} \tilde{T}_i^{-\alpha}}.$$

   (d) Compress $K$ (e.g., using Elias delta coding [29]) and sends $K$.

3. The decoder, which knows $(Z_i)_i, K$, outputs $Z = Z_K$.

Note that when $\alpha = \infty$, we have $K = \operatorname{argmin}_i \tilde{T}_i$, and PPR reduces to the original Poisson functional representation [61, 60]. PPR can simulate the privacy mechanism $P_{Z|X}$ precisely, as shown in the following proposition. The proof is in Appendix A.

**Proposition 4.2.** *The output $Z$ of PPR follows the conditional distribution $P_{Z|X}$ exactly.*

Due to the *exactness* of PPR, it guarantees unbiasedness for tasks such as DME. If the goal is only to design a stand-alone privacy mechanism, we can focus on the privacy and utility of the mechanism without studying the output distribution. However, if the output of the mechanism is used for downstream tasks (e.g., for DME, after receiving information from clients, the server sends information about the aggregated mean to data analysts, where central DP is crucial), having an exact characterization of the conditional distribution of the output given the input allows us to obtain precise (central) privacy and utility guarantees.

Notably, PPR is *universal* in the sense that only the encoder needs to know the simulated mechanism $P_{Z|X}$. The decoder can decode the index $K$ as long as it has access to the shared randomness $(Z_i)_i$. This allows the encoder to choose an arbitrary mechanism $P_{Z|X}$ with the same $\mathcal{Z}$, and adapt the choice of $P_{Z|X}$ to the communication and privacy constraints without explicitly informing the decoder which mechanism is chosen.

Practically, the algorithm cannot compute the whole infinite sequence $(\tilde{T}_i)_i$. We can truncate the method and only compute $\tilde{T}_i, \ldots, \tilde{T}_N$ for a large $N$ and select $K \in \{1, \ldots, N\}$, which incurs a small distortion in the distribution of $Z$.[7] While this method is practically acceptable, it might defeat

---

[5]The original Poisson functional representation [61, 60] uses the whole $(Z_i, T_i)_i$ as the shared randomness. It is clear that $(T_i)_i$ is not needed by the decoder, and hence we can use only $(Z_i)_i$ as the shared randomness.

[6]We note that our analyses assume that the adversary knows both the index $K$ and the shared randomness $(Z_i)_i$, and we prove that the mechanism is still private despite the shared randomness between the encoder and the decoder, since the privacy is provided by locally randomizing $K$ in Step 2c.

[7]To compare to the minimal random coding (MRC) [47, 20, 74] scheme in [71], which also utilizes a finite number $N$ of samples $(Z_i)_{i=1,\ldots,N}$, while truncating the number of samples to $N$ in both PPR and MRC

the purpose of having an exact algorithm that ensures the correct conditional distribution $P_{Z|X}$. In Appendix B, we will present an exact algorithm for PPR that terminates in a finite amount of time, using a reparametrization that allows the encoder to know when the optimal point $Z_i$ has already been encountered (see Algorithm 1 in Appendix B).

By the lower bound for channel simulation [6, 61], we must have $H(K) \geq I(X; Z)$, i.e., the compression size is at least the mutual information between the data $X$ and the output $Z$. The following result shows that the compression provided by PPR is "almost optimal", i.e., close to the theoretical lower bound $I(X; Z)$. The proof is given in Appendix F.

**Theorem 4.3** (Compression size of PPR). *For PPR with parameter $\alpha > 1$, when the encoder is given the input $x$, the message $K$ given by PPR satisfies*

$$\mathbb{E}[\log K] \leq D(P \| Q) + (\log(3.56))/\min\{(\alpha - 1)/2, 1\},$$

*where $P := P_{Z|X}(\cdot | x)$. As a result, when the input $X \sim P_X$ is random, taking $Q = P_Z$, we have*

$$\mathbb{E}[\log K] \leq I(X; Z) + (\log(3.56))/\min\{(\alpha - 1)/2, 1\}.$$

Note the running time complexity (which depends on the number of samples $Z_i$ the algorithm must examine before outputting the index $K$) can be quite high. Since $\mathbb{E}[\log K] \approx I(X; Z)$, $K$ (and hence the running time) is at least exponential in $I(X; Z)$. See more discussions in Section 8.

If a prefix-free encoding of $K$ is required, then the number of bits needed is slightly larger than $\log_2 K$. For example, if Elias delta code [29] is used, the expected compression size is $\leq \mathbb{E}[\log_2 K] + 2 \log_2(\mathbb{E}[\log_2 K] + 1) + 1$ bits. If the Shannon code [73] (an almost-optimal prefix-free code) for the Zipf distribution $p(k) \propto k^{-\lambda}$ with $\lambda = 1 + 1/\mathbb{E}[\log_2 K]$ is used, the expected compression size is $\leq \mathbb{E}[\log_2 K] + \log_2(\mathbb{E}[\log_2 K] + 1) + 2$ bits (see [61]). Both codes yield an $I(X; Z) + O(\log I(X; Z))$ size, within a logarithmic gap from the lower bound $I(X; Z)$. This is similar to some other channel simulation schemes such as [45, 10, 61], though these schemes do not provide privacy guarantees.

Note that if $P_{Z|X}$ is $\varepsilon$-DP, then by definition, for any $z \in \mathcal{Z}$ and $x, x_0 \in \mathcal{X}$, it holds that

$$D\left(P_{Z|X=x} \middle\| P_{Z|X=x_0}\right) = \mathbb{E}_{Z \sim P_{Z|X=x}}\left[\log\left(\frac{\mathrm{d}P_{Z|X=x}}{\mathrm{d}P_{Z|X=x_0}}(Z)\right)\right] \leq \varepsilon \log e.$$

Setting the proposal distribution $Q = P_{Z|X=x_0}$ for an arbitrary $x_0 \in \mathcal{X}$ gives the following bound.

**Corollary 4.4** (Compression size under $\varepsilon$-LDP). *Let $P_{Z|X}$ satisfy $\varepsilon$-differential privacy. Let $x_0 \in \mathcal{X}$ and $Q = P_{Z|X=x_0}$. Then for PPR with parameter $\alpha > 1$, the expected compression size is at most $\ell + \log_2(\ell + 1) + 2$ bits, where $\ell := \varepsilon \log_2 e + (\log_2(3.56))/\min\{(\alpha - 1)/2, 1\}$.*

Next, we analyze the privacy guarantee of PPR. The PPR method induces a conditional distribution $P_{(Z_i)_i, K|X}$ of the knowledge of the decoder $((Z_i)_i, K)$, given the data $X$. To analyze the privacy guarantee, we study whether the randomized mapping $P_{(Z_i)_i, K|X}$ from $X$ to $((Z_i)_i, K)$ satisfies $\varepsilon$-DP or $(\varepsilon, \delta)$-DP.[8] This is similar to the privacy condition in [71], and is referred as *decoder privacy* in [72], which is stronger than *database privacy* which concerns the privacy of the randomized mapping from $X$ to the final output $Z$ [72] (which is simply the privacy of the original mechanism $P_{Z|X}$ to be compressed since PPR simulates $P_{Z|X}$ precisely). Since the decoder knows $((Z_i)_i, K)$, more than just the final output $Z$, we expect that the PPR-compressed mechanism $P_{(Z_i)_i, K|X}$ to have a worse privacy guarantee than the original mechanism $P_{Z|X}$, which is the price of having a smaller communication cost. The following result shows that, if the original mechanism $P_{Z|X}$ is $\varepsilon$-DP, then the PPR-compressed mechanism is guaranteed to be $2\alpha\varepsilon$-DP.

---

results in a distortion in the distribution of $Z$ that tends to 0 as $N \to \infty$, the difference is that $\log K$ (which is approximately the compression size) in MRC grows like $\log N$, whereas $\log K$ does not grow as $N \to \infty$ in PPR. The size $N$ in truncated PPR merely controls the tradeoff between accuracy of the distribution of $Z$ and the running time of the algorithm.

[8]Note that the encoder does not actually send $((Z_i)_i, K)$; it only sends $K$. The common randomness $(Z_i)_i$ is independent of the data $X$, and can be pre-generated using a common random seed in practice. While this seed must be communicated between the client and the server as a small overhead, the client and the server only ever need to communicate *one* seed to initialize a pseudorandom number generator, that can be used in *all* subsequent privacy mechanisms and communication tasks (to transmit high-dimensional data or use DP mechanisms for many times). The conditional distribution $P_{(Z_i)_i, K|X}$ is only relevant for privacy analysis.

**Theorem 4.5** ($\varepsilon$-DP of PPR). *If the mechanism $P_{Z|X}$ is $\varepsilon$-differentially private, then PPR $P_{(Z_i)_i, K|X}$ with parameter $\alpha > 1$ is $2\alpha\varepsilon$-differentially private.*

Similar results also apply to $(\varepsilon, \delta)$-DP and metric DP.

**Theorem 4.6** (($\varepsilon, \delta$)-DP of PPR). *If the mechanism $P_{Z|X}$ is $(\varepsilon, \delta)$-differentially private, then PPR $P_{(Z_i)_i, K|X}$ with parameter $\alpha > 1$ is $(2\alpha\varepsilon, 2\delta)$-differentially private.*

**Theorem 4.7** (Metric privacy of PPR). *If the mechanism $P_{Z|X}$ satisfies $\varepsilon \cdot d_{\mathcal{X}}$-privacy, then PPR $P_{(Z_i)_i, K|X}$ with parameter $\alpha > 1$ satisfies $2\alpha\varepsilon \cdot d_{\mathcal{X}}$-privacy.*

Refer to Appendices C and D for the proofs. In Theorem 4.5 and Theorem 4.6, PPR imposes a multiplicative penalty $2\alpha$ on the privacy parameter $\varepsilon$. This penalty can be made arbitrarily close to 2 by taking $\alpha$ close to 1, which increases the communication cost (see Theorem 4.3). Compared to minimal random coding which has a factor 2 penalty in the DP guarantee [47, 71], the $2\alpha$ factor in PPR is slightly larger, though PPR ensures exact simulation (unlike [47, 71] which are approximate). The method in [31] does not have a penalty on $\varepsilon$, but the utility and compression size depends on computational hardness assumptions on the pseudorandom number generator, and there is no guarantee that the compression size is close to the optimum. In comparison, the compression and privacy guarantees of PPR are *unconditional* and does not rely on computational assumptions.

In order to make the penalty of PPR close to 1, we have to consider $(\varepsilon, \delta)$-differential privacy, and allow a small failure probability, i.e., a small increase in $\delta$. The following result shows that PPR can compress any $\varepsilon$-DP mechanism into a $(\approx \varepsilon, \approx 0)$-DP mechanism as long as $\alpha$ is close enough to 1 (i.e., almost no inflation). More generally, PPR can compress an $(\varepsilon, \delta)$-DP mechanism into an $(\approx \varepsilon, \approx 2\delta)$-DP mechanism for $\alpha$ close to 1. The proof is in Appendix E.

**Theorem 4.8** (Tighter $(\varepsilon, \delta)$-DP of PPR). *If the mechanism $P_{Z|X}$ is $(\varepsilon, \delta)$-differentially private, then PPR $P_{(Z_i)_i, K|X}$ with parameter $\alpha > 1$ is $(\alpha\varepsilon + \tilde{\varepsilon}, 2(\delta + \tilde{\delta}))$-differentially private, for every $\tilde{\varepsilon} \in (0, 1]$ and $\tilde{\delta} \in (0, 1/3]$ that satisfy $\alpha \leq e^{-4.2}\tilde{\delta}\tilde{\varepsilon}^2/(-\ln\tilde{\delta}) + 1$.*

## 5 Applications to Distributed Mean Estimation

We demonstrate the efficacy of PPR by applying it to distributed mean estimation (DME) [75]. Note that private DME is the core sub-routine in various private and federated optimization algorithms, such as DP-SGD [1] or DP-FedAvg [67].

Consider the following general distributed setting: each of $n$ clients holds a local data point $X_i \in \mathcal{X}$, and a central server aims to estimate a function of all local data $\mu(X^n)$, subject to privacy and local communication constraints. To this end, each client $i$ compresses $X_i$ into a message $Z_i \in \mathcal{Z}_n$ via a local encoder, and we require that each $Z_i$ can be encoded into a bit string with an expected length of at most $b$ bits. Upon receiving $Z^n := (Z_1, \ldots, Z_n)$, the central server decodes it and outputs a DP estimate $\hat{\mu}$. Two DP criteria can be considered: the $(\varepsilon, \delta)$-central DP of the randomized mapping from $X^n$ to $\hat{\mu}$, and the $(\varepsilon, \delta)$-local DP of the randomized mapping from $X_i$ to $Z_i$ for each client $i$.

In the distributed $L_2$ mean estimation problem, $\mathcal{X} = \mathcal{B}_d(C) := \{v \in \mathbb{R}^d \mid \|v\|_2 \leq C\}$, and the central server aims to estimate the sample mean $\mu(X^n) := \frac{1}{n}\sum_{i=1}^n X_i$ by minimizing the mean squared error (MSE) $\mathbb{E}[\|\mu - \hat{\mu}\|_2^2]$. It is recently proved that under $\varepsilon$-local DP, privUnit [8, 4] is the optimal mechanism. By simulating privUnit with PPR and applying Corollary 4.4 and Theorem 4.6, we immediately obtain the following corollary:

**Corollary 5.1** (PPR simulating privUnit). *Let $P$ be the density defined by $\varepsilon$-privUnit$_2$ Bhowmick et al. [8, Algorithm 1]. Let $Q$ be the uniform density over the sphere $\mathbb{S}^{d-1}(1/m)$ where the radius $1/m$ is defined in Bhowmick et al. [8, (15)]. Let $r^* := e^\varepsilon$. Then the outcome of PPR (see Algorithm 1) satisfies (1) $2\alpha\varepsilon$-local DP; and (2) $(\alpha\varepsilon + \tilde{\varepsilon}, 2\tilde{\delta})$-DP for any $\alpha \leq e^{-4.2}\tilde{\delta}\tilde{\varepsilon}^2/\log(1/\tilde{\delta}) + 1$. In addition, the average compression size is at most $\ell + \log_2(\ell + 1) + 2$ bits where $\ell := \varepsilon + (\log_2(3.56))/\min\{(\alpha - 1)/2, 1\}$. Moreover, PPR achieves the same MSE as $\varepsilon$-privUnit$_2$, which is $O\left(d/\min\left(\varepsilon, \varepsilon^2\right)\right)$.*

Note that PPR can simulate arbitrary local DP mechanisms. However, we present only the result of privUnit$_2$ because it achieves the optimal privacy-accuracy trade-off. Besides simulating local DP mechanisms, PPR can also compress central DP mechanisms while still preserving some (albeit weaker) local guarantees. We give a corollary of Theorems 4.3 and 4.6. The proof is in Appendix H.

**Corollary 5.2** (PPR-compressed Gaussian mechanism). *Let $\varepsilon, \delta \in (0, 1)$. Consider the Gaussian mechanism $P_{Z|X}(\cdot|x) = \mathcal{N}(x, \frac{\sigma^2}{n}\mathbb{I}_d)$, and the proposal distribution $Q = \mathcal{N}(0, (\frac{C^2}{d} + \frac{\sigma^2}{n})\mathbb{I}_d)$, where $\sigma \geq \frac{C\sqrt{2\ln(1.25/\delta)}}{\varepsilon}$. For each client $i$, let $Z_i$ be the output of PPR applied on $P_{Z|X}(\cdot|X_i)$. We have:*

- *$\hat{\mu}(Z^n) := \frac{1}{n}\sum_i Z_i$ yields an unbiased estimator of $\mu(X^n) = \frac{1}{n}\sum_{i=1}^n X_i$ satisfying $(\varepsilon, \delta)$-central DP and has MSE $\mathbb{E}[\|\mu - \hat{\mu}\|_2^2] = \sigma^2 d/n^2$.*

- *As long as $\varepsilon < 1/\sqrt{n}$, PPR satisfies $(2\alpha\sqrt{n}\varepsilon, 2\delta)$-local DP.[9]*

- *The average per-client communication cost is at most $\ell + \log_2(\ell + 1) + 2$ bits where*

$$\ell := \frac{d}{2}\log_2\left(\frac{C^2 n}{d\sigma^2} + 1\right) + \eta_\alpha \leq \frac{d}{2}\log_2\left(\frac{n\varepsilon^2}{2d\ln(1.25/\delta)} + 1\right) + \eta_\alpha,$$

*where $\eta_\alpha := (\log_2(3.56))/\min\{(\alpha - 1)/2, 1\}$.*

A few remarks are in order. First, notice that when $\alpha$ is fixed, for an $O(\frac{C^2 d}{n^2\varepsilon^2}\log(1/\delta))$ MSE, the per-client communication cost is

$$O\left(d\log\left(\frac{n\varepsilon^2}{d\log(1/\delta)} + 1\right) + 1\right),$$

which is at least as good as the $O(n\varepsilon^2/\log(1/\delta) + 1)$ bound in [75, 19], and can be better than $O(n\varepsilon^2/\log(1/\delta) + 1)$ when $n \gg d$. Hence, the PPR-compressed Gaussian mechanism is order-wise optimal. Second, compared to other works that also compress the Gaussian mechanism, PPR is the only lossless compressor; schemes based on random sparsification, projection, or minimum random coding (e.g., Triastcyn et al. [78], Chen et al. [19]) are *lossy*, i.e., they introduce additional distortion on top of the DP noise. Finally, other DP mechanism compressors tailored to local randomizers [31, 71] do not provide the same level of central DP guarantees when applied to local Gaussian noise since the reconstructed noise is no longer Gaussian. Refer to Section 7 for experiments.

## 6 Applications to Metric Privacy

Metric privacy [16, 3] (see Definition 3.2) allows users to send privatized version $Z \in \mathbb{R}^d$ of their data vectors $X \in \mathbb{R}^d$ to an untrusted server, so that the server can know $X$ approximately but not exactly. A popular mechanism is the *Laplace mechanism* [16, 3, 33, 34], where a $d$-dimensional Laplace noise is added to $X$. The conditional density function of $Z$ given $X$ is $f_{Z|X}(z|x) \propto e^{-\varepsilon d_\mathcal{X}(x,z)}$, where $\varepsilon$ is the privacy parameter, and the metric $d_\mathcal{X}(x, z) = \|x - z\|_2$ is the Euclidean distance. The Laplace mechanism achieves $\varepsilon \cdot d_\mathcal{X}$-privacy, and has been used, for example, in geo-indistinguishability to privatize the users' locations [3], and to privatize high-dimensional word embedding vectors [33, 34].

A problem is that the real vector $Z$ cannot be encoded into finitely many bits. To this end, [3] studies a *discrete Laplace mechanism* where each coordinate of $Z$ is quantized to a finite number of levels, introducing additional distortion to $Z$. PPR provides an alternative compression method that preserves the statistical behavior of $Z$ (e.g., unbiasedness) exactly. We give a corollary of Theorems 4.3 and 4.7. The proof is in Appendix I. Refer to Appendix J for an experiment on metric privacy.

**Corollary 6.1** (PPR-compressed Laplace mechanism). *Consider PPR applied to the Laplace mechanism $P_{Z|X}$ where $X \in \mathcal{B}_d(C) = \{x \in \mathbb{R}^d \mid \|x\|_2 \leq C\}$, with a proposal distribution $Q = \mathcal{N}(0, (\frac{C^2}{d} + \frac{d+1}{\varepsilon^2})\mathbb{I}_d)$. It achieves an MSE $\frac{d(d+1)}{\varepsilon^2}$, a $2\alpha\epsilon \cdot d_\mathcal{X}$-privacy, and a compression size at most $\ell + \log_2(\ell + 1) + 2$ bits, where*

$$\ell := \frac{d}{2}\log_2\left(\frac{2}{e}\left(\frac{C^2\varepsilon^2}{d} + d + 1\right)\right) - \log_2\frac{\Gamma(d+1)}{\Gamma(\frac{d}{2}+1)} + \eta_\alpha,$$

*where $\eta_\alpha := (\log_2(3.56))/\min\{(\alpha - 1)/2, 1\}$.*

---

[9]The restricted range on $\varepsilon < 1/\sqrt{n}$ is due to the simpler privacy accountant [25]. By using the Rényi DP accountant instead, one can achieve a tighter result that applies to any $n$. We present the Rényi DP version of the corollary in Appendix G. Moreover, in the context of federated learning, $n$ refers to the number of clients in *each round*, which is typically much smaller than the total number of clients. For example, as observed in [52], the per-round cohort size in Google's FL application typically ranges from $10^3$ to $10^5$, significantly smaller than the number of trainable parameters $d \in [10^6, 10^9]$ or the number of available users $N \in [10^6, 10^8]$.

# 7 Empirical Results

We empirically evaluate our scheme on the DME problem (which is formally introduced in Section 5), examine the privacy-accuracy-communication trade-off, and compare it with the Coordinate Subsampled Gaussian Mechanism (CSGM) [19, Algorithm 1], an order-optimal scheme for DME under central DP. In Chen et al. [19], each client only communicates partial information (via sampling a subset of the coordinates of the data vector) about its samples to amplify the privacy, and the compression is mainly from subsampling. Moreover, CSGM only guarantees central DP.

We use the same setup that has been used in [19]: consider $n = 500$ clients, and the dimension of local vectors is $d = 1000$, each of which is generated according to $X_i(j) \overset{\text{i.i.d.}}{\sim} (2 \cdot \text{Ber}(0.8) - 1)$, where $\text{Ber}(0.8)$ is a Bernoulli random variable with parameter $p = 0.8$. We require $(\varepsilon, \delta)$-central DP with $\delta = 10^{-6}$ and $\varepsilon \in [0.05, 6]$ and apply the PPR with $\alpha = 2$ to simulate the Gaussian mechanism, where the privacy budgets are accounted via Rényi DP.

We compare the MSE of PPR ($\alpha = 2$, using Theorem 4.3) and CSGM under various compression sizes in Figure 1 (the $y$-axis is in logarithmic scale).[10] Note that the MSE of the (uncompressed) Gaussian mechanism coincides with the CSGM with 1000 bits, and the PPR with only 400 bits. We see that PPR consistently achieves a smaller MSE compared to CSGM for all $\varepsilon$'s and compression sizes considered. For $\epsilon = 1$ and we compress $d = 1000$ to 50 bits, CSGM has an MSE 0.1231 , while PPR has an MSE 0.08173, giving a 33.61% reduction. For $\epsilon = 0.5$ and we compress $d = 1000$ to 25 bits (the case of high compression and conservative privacy), CSGM has an MSE 0.3877, while PPR has an MSE 0.3011, giving a 22.33% reduction. These reductions are significant, since all considered mechanisms are asymptotically close to optimal and a large improvement compared to an (almost optimal) mechanism is unexpected. See Section L for more about MSE against the compression sizes.

We also emphasize that PPR provides *both* central and local DP guarantees according to Theorem 4.5, 4.6 and 4.8. In contrast, CSGM only provides central DP guarantees. Another advantage of PPR under conservative privacy (small $\epsilon$) is that the trade-off between $\epsilon$ and MSE of PPR exactly coincides with the trade-off of the Gaussian mechanism for small $\epsilon$ (see Figure 1), and CSGM is only close to (but strictly worse than) the Gaussian mechanism. This means that for small $\epsilon$, PPR provides compression without any drawback in terms of $\epsilon$-MSE trade-off compared to the Gaussian mechanism (which requires an infinite size communication to exactly realize).

Moreover, although directly applying PPR on the $d$-dimensional vectors is impractical for a large $d$, one can ensure an efficient $O(d)$ running time (see Section 8 for details) by breaking the vector with $d = 1000$ dimensions into small chunks of fixed lengths (we use $d_{\text{chunk}} = 50$ dimensions for each chunk), and apply the PPR to each chunk. We call it the *sliced PPR* in Figure 1. Though the sliced PPR has a small penalty on the MSE (as shown in Figure 1), it still outperforms the CSGM (400 bits) for the range of $\varepsilon$ in the plot. For the sliced PPR for one $d = 1000$ vector, when $\epsilon = 0.05$, the running time is 1.3348 seconds on average.[11] For larger $\epsilon$'s, we can choose smaller $d_{\text{chunk}}$'s to have reasonable running time: For $\epsilon = 6$ and $d_{\text{chunk}} = 2$ we have an average running time 0.0127 seconds and with $d_{\text{chunk}} = 4$ we have an average running time 0.6343 seconds; for $\epsilon = 10$ and $d_{\text{chunk}} = 2$ we have an average running time 0.0128 seconds and with $d_{\text{chunk}} = 4$ we have an average running time 0.7301 seconds. See Appendix K for more experiments on the running time of the sliced PPR.

# 8 Limitations

While PPR is communication-efficient, having only a logarithmic gap from the theoretical lower bound on the compression size as shown in Theorem 4.3, the running time complexity can be high. However, we note that an exponential complexity is also needed in sampling methods that do not ensure privacy, such as [65, 47]. It has been proved in [2] that no polynomial time general sampling-

---

[10]Source code: `https://github.com/cheuktingli/PoissonPrivateRepr`. Experiments were executed on M1 Pro Macbook, 8-core CPU ($\approx$ 3.2 GHz) with 16GB memory. For PPR under a privacy budget $\varepsilon$ and communication budget $b$, we find the largest $\varepsilon' \leq \varepsilon$ such that the communication cost bound in Theorem 4.3 (with Shannon code [73]) for simulating the Gaussian mechanism with $(\varepsilon', \delta)$-central DP is at most $b$, and use PPR to simulate this Gaussian mechanism. Thus, MSE of PPR in Figure 1 becomes flat for large $\varepsilon$, as PPR falls back to using a smaller $\varepsilon'$ instead of $\varepsilon$ due to the communication budget.

[11]The running time is calculated by $\frac{1000}{50} \times T_{\text{chunk}}$, where each chunk's running time $T_{\text{chunk}}$ is averaged over 1000 trials. The estimate of the mean of $T_{\text{chunk}}$ is 0.0667, whereas the standard deviation is 0.2038.

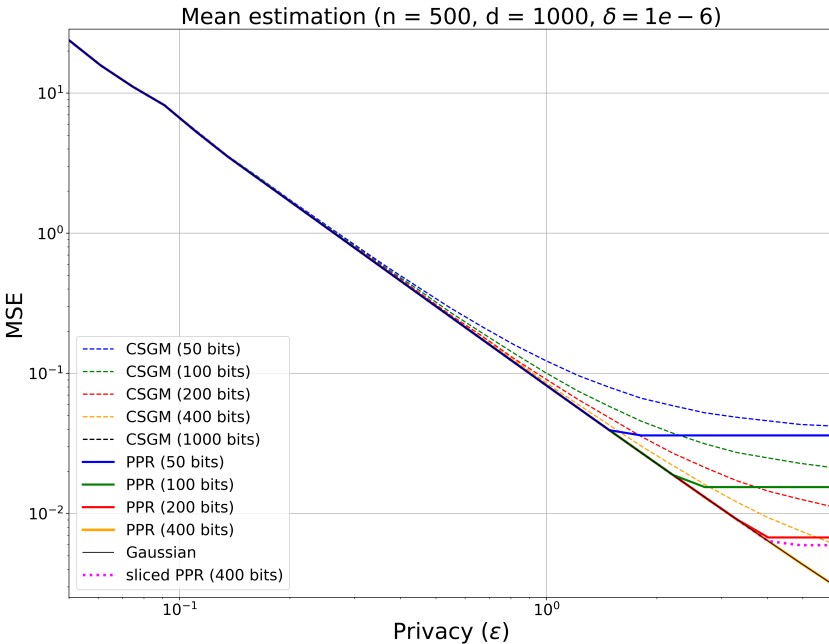

Figure 1: MSE of distributed mean estimation for PPR and CSGM [19] for different $\varepsilon$'s.

based method exists (even without privacy constraint), if $RP \neq NP$. All existing polynomial time exact channel simulation methods can only simulate specific noisy channels.[12] Hence, a polynomial time algorithm for exactly compressing a general DP mechanism is likely nonexistent.

Nevertheless, this is not an obstacle for simulating local DP mechanisms, since the mutual information $I(X; Z)$ for a reasonable local DP mechanism must be small, or else the leakage of the data $X$ in $Z$ would be too large. For an $\varepsilon$-local DP mechanism, we have $I(X; Z) \leq \min\{\varepsilon, \varepsilon^2\}$ (in nats) [22]. Hence, the PPR algorithm can terminate quickly even if has a running time exponential in $I(X; Z)$.

Another way to ensure a polynomial running time is to divide the data into small chunks and apply the mechanism to each chunk separately. For example, to apply the Gaussian mechanism to a high-dimensional vector, we break it into several shorter vectors and apply the mechanism to each vector. Experiments in Section 7 show that this greatly reduces the running time while having only a small penalty on the compression size. See Appendix K for experiments on the running time of PPR.

## 9 Conclusion

We proposed a novel scheme for compressing DP mechanisms, called Poisson private representation (PPR). Unlike previous schemes which are either constrained on special classes of DP mechanisms or introducing additional distortions on the output, our scheme can compress and exactly simulate arbitrary mechanisms while protecting differential privacy, with a compression size that is close to the theoretic lower bound. A future direction is to reduce the running time of PPR under certain restrictions on $P_{Z|X}$. For example, the techniques in [38, 35] may be useful when $P_{Z|X}$ is unimodal.

### Acknowledgment

YL was partially supported by the CUHK PhD International Mobility for Partnerships and Collaborations Award 2023-24. WC and AO were supported by the NSF grant CIF-2213223. CTL was partially supported by two grants from the Research Grants Council of the Hong Kong Special Administrative Region, China [Project No.s: CUHK 24205621 (ECS), CUHK 14209823 (GRF)].

---

[12]For example, [36] and dithered-quantization-based schemes [48, 72] can only simulate additive noise mechanisms. Among these existing works, only [72] ensures local DP.

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

# A  Proof of Proposition 4.2

Write $(X_i)_i \sim \mathrm{PP}(\mu)$ if the points $(X_i)_i$ (as a multiset, ignoring the ordering) form a Poisson point process with intensity measure $\mu$. Similarly, for $f : [0, \infty)^n \to [0, \infty)$, we write $\mathrm{PP}(f)$ for the Poisson point process with intensity function $f$ (i.e., the intensity measure has a Radon-Nikodym derivative $f$ against the Lebesgue measure).

Let $(T_i)_i \sim \mathrm{PP}(1)$ be a Poisson process with rate 1, independent of $Z_1, Z_2, \ldots \overset{iid}{\sim} Q$. By the marking theorem [57], $(Z_i, T_i)_i \sim \mathrm{PP}(Q \times \lambda_{[0,\infty)})$, where $Q \times \lambda_{[0,\infty)}$ is the product measure between $Q$ and the Lebesgues measure over $[0, \infty)$. Let $P = P_{Z|X}(\cdot|x)$, and $\tilde{T}_i = T_i \cdot (\frac{\mathrm{d}P}{\mathrm{d}Q}(Z_i))^{-1}$. By the mapping theorem [57] (also see [61, 60]), $(Z_i, \tilde{T}_i)_i \sim \mathrm{PP}(P \times \lambda_{[0,\infty)})$. Note that the points $(Z_i, \tilde{T}_i)_i$ may not be sorted in ascending order of $\tilde{T}_i$. Therefore, we will sort them as follows. Let $j_1$ be the $j$ such that $\tilde{T}_j$ is the smallest, $j_2$ be the $j$ other than $j_1$ such that $\tilde{T}_j$ is the smallest, and so on. Break ties arbitrarily. Then $(\tilde{T}_{j_i})_i$ is an ascending sequence, and we still have $(Z_{j_i}, \tilde{T}_{j_i})_i \sim \mathrm{PP}(P \times \lambda_{[0,\infty)})$ since we are merely rearranging the points. Comparing $(Z_{j_i}, \tilde{T}_{j_i})_i \sim \mathrm{PP}(P \times \lambda_{[0,\infty)})$ with the definition of $(Z_i, T_i)_i \sim \mathrm{PP}(Q \times \lambda_{[0,\infty)})$, we can see that $(\tilde{T}_{j_i})_i \sim \mathrm{PP}(1)$ is independent of $Z_{j_1}, Z_{j_2}, \ldots \overset{iid}{\sim} P$. Recall that in PPR, we generate $K \in \mathbb{Z}_+$ with

$$\Pr(K = k) = \frac{\tilde{T}_k^{-\alpha}}{\sum_{i=1}^{\infty} \tilde{T}_i^{-\alpha}},$$

and the final output is $Z_K$. Rearranging the points according to $(j_i)_i$, the distribution of the final output remains the same if we instead generate $K' \in \mathbb{Z}_+$ with

$$\Pr(K' = k) = \frac{\tilde{T}_{j_k}^{-\alpha}}{\sum_{i=1}^{\infty} \tilde{T}_{j_i}^{-\alpha}},$$

and the final output is $Z_{j_{K'}}$. Since $(\tilde{T}_{j_i})_i \sim \mathrm{PP}(1)$ is independent of $Z_{j_i} \overset{iid}{\sim} P$, we know that $K'$ is independent of $(Z_{j_i})_i$, and hence $Z_{j_{K'}} \sim P$ follows the desired distribution.

# B  Reparametrization and Detailed Algorithm of PPR

We now discuss the implementation of the Poisson private representation in Section 4. Practically, the algorithm cannot compute the whole infinite sequence $(\tilde{T}_i)_i$. We now present an exact algorithm for PPR that terminates in a finite amount of time using a reparametrization.

In the proof of Theorem F.1, we showed that, letting $(T_i)_i \sim \mathrm{PP}(1)$, $Z_1, Z_2, \ldots \overset{iid}{\sim} Q$, $R_i := (\mathrm{d}P/\mathrm{d}Q)(Z_i)$, $V_1, V_2, \ldots \overset{iid}{\sim} \mathrm{Exp}(1)$, PPR can be equivalently expressed as

$$K = \underset{k}{\mathrm{argmin}}\, T_k^{\alpha} R_k^{-\alpha} V_k.$$

The problem of finding $K$ is that there is no stopping criteria for the argmin. For example, if we scan the points $(T_i, R_i, V_i)_i$ in increasing order of $T_i$, it is always possible that there is a future point with $V_i$ so small that it makes $T_i^{\alpha} R_i^{-\alpha} V_i$ smaller than the current minimum. If we scan the points in increasing order of $V_i$ instead, it is likewise possible that there is a future point with a very small $T_i$. We can scan the points in increasing order of $U_i := T_i^{\alpha} V_i$, but we would not know the indices of the points in the original process where $T_1 \leq T_2 \leq \cdots$ is in increasing order, which is necessary to find out the $Z_i$ corresponding to each point (recall that in PPR, the point with the smallest $T_i$ corresponds to $Z_1$, the second smallest $T_i$ corresponds to $Z_2$, etc.).

Therefore, we will scan the points in increasing order of $B_i := T_i^{\alpha} \min\{V_i, 1\}$ instead. By the mapping theorem [57], $(T_i^{\alpha})_i \sim \mathrm{PP}(\alpha^{-1} t^{1/\alpha - 1})$. By the marking theorem [57],

$$(T_i^{\alpha}, V_i)_i \sim \mathrm{PP}(\alpha^{-1} t^{1/\alpha - 1} e^{-v}).$$

By the mapping theorem,

$$(T_i^{\alpha}, T_i^{\alpha} V_i)_i \sim \mathrm{PP}(\alpha^{-1} t^{1/\alpha - 2} e^{-vt^{-1}}).$$

Since $B_i = \min\{T_i^\alpha, T_i^\alpha V_i\}$, again by the mapping theorem,

$$(B_i)_i \sim \mathrm{PP}\bigg( \int_b^\infty \alpha^{-1} b^{1/\alpha-2} e^{-vb^{-1}} \mathrm{d}v$$

$$+ \int_b^\infty \alpha^{-1} t^{1/\alpha-2} e^{-bt^{-1}} \mathrm{d}t \bigg)$$

$$= \mathrm{PP}\left( \alpha^{-1} b^{1/\alpha-1} e^{-1} + \alpha^{-1} b^{1/\alpha-1} \gamma(1-\alpha^{-1}, 1) \right)$$

$$= \mathrm{PP}\left( \alpha^{-1} \left( e^{-1} + \gamma_1 \right) b^{1/\alpha-1} \right),$$

where $\gamma_1 := \gamma(1-\alpha^{-1}, 1)$ and $\gamma(\beta, x) = \int_0^x e^{-\tau} \tau^{\beta-1} \mathrm{d}\tau$ is the lower incomplete gamma function. Comparing the distribution of $(B_i)_i$ and $(T_i^\alpha)_i$, we can generate $(B_i)_i$ by first generating $(U_i)_i \sim \mathrm{PP}(1)$, and then taking $B_i = (U_i \alpha/(e^{-1} + \gamma_1))^\alpha$. The conditional distribution of $(T_i, V_i)$ given $B_i = b$ is described as follows:

- With probability $e^{-1}/(e^{-1} + \gamma_1)$, we have $T_i^\alpha = b$ and $T_i^\alpha V_i \sim b(\mathrm{Exp}(1) + 1)$, and hence $T_i = b^{1/\alpha}$ and $V_i \sim \mathrm{Exp}(1) + 1$.
- With probability $\gamma_1/(e^{-1} + \gamma_1)$, we have $T_i^\alpha V_i = b$ and

$$T_i^\alpha \sim \frac{\alpha^{-1} t^{1/\alpha-2} e^{-bt^{-1}}}{\alpha^{-1} \gamma(1-\alpha^{-1}, 1) b^{1/\alpha-1}}.$$

Hence, for $0 < \tau \le 1$,

$$\Pr(V_i \le \tau) = \Pr(T_i^\alpha \ge b/\tau) = \frac{\gamma(1-\alpha^{-1}, \tau)}{\gamma(1-\alpha^{-1}, 1)},$$

and $V_i$ follows the truncated gamma distribution with shape $1 - \alpha^{-1}$ and scale 1, truncated within the interval $[0, 1]$. We then have $T_i = (b/V_i)^{1/\alpha}$.

The algorithm is given in Algorithm 1. The encoder and decoder require a shared random seed $s$. One way to generate $s$ is to have the encoder and decoder maintain two synchronized pseudorandom number generators (PRNGs) that are always at the same state, and invoke the PRNGs to generate $s$, guaranteeing that the $s$ at the encoder is the same as the $s$ at the decoder. The encoder maintains a collection of points $(T_i, V_i, \Theta_i)$, stored in a heap to allow fast query and removal of the point with the smallest $T_i$. The value $\Theta_i \in \{0, 1\}$ indicates whether it is possible that the point $(T_i, V_i)$ attains the minimum of $T_k^\alpha R_k^{-\alpha} V_k$. The encoding algorithm repeats until there is no possible points left in the heap, and it is impossible for any future point to be better than the current minimum of $T_k^\alpha R_k^{-\alpha} V_k$. The encoding time complexity is $O(\sup_z (\mathrm{d}P/\mathrm{d}Q)(z) \log(\sup_z (\mathrm{d}P/\mathrm{d}Q)(z)))$, which is close to other sampling-based channel simulation schemes [45, 36].[13] The decoding algorithm simply outputs the $k$-th sample generated using the random seed $s$, which can be performed in $O(1)$ time.[14]

The PPR is implemented by Algorithm 1. We write $x \leftarrow \mathrm{Exp}_{\mathscr{G}}(1)$ to mean that we generate an exponential random variate $x$ with rate 1 using the pseudorandom number generator $\mathscr{G}$. Write $x \leftarrow \mathrm{Exp}_{\mathrm{local}}(1)$ to mean that $x$ is generated using a local pseudorandom number generator (not $\mathscr{G}$).

## C  Proofs of Theorem 4.5 and Theorem 4.7

First prove Theorem 4.5. Consider a $\varepsilon$-DP mechanism $P_{Z|X}$. Consider neighbors $x_1, x_2$, and let $P_j := P_{Z|X}(\cdot|x_j)$, $\tilde{T}_{j,i} := T_i/(\frac{\mathrm{d}P_j}{\mathrm{d}Q}(Z_i))$, and $K_j$ be the output of PPR applied on $P_j$, for $j = 1, 2$. Since $P_{Z|X}$ is $\varepsilon$-DP,

$$e^{-\varepsilon} \frac{\mathrm{d}P_2}{\mathrm{d}Q}(z) \le \frac{\mathrm{d}P_1}{\mathrm{d}Q}(z) \le e^\varepsilon \frac{\mathrm{d}P_2}{\mathrm{d}Q}(z) \tag{2}$$

---

[13]It was shown in [36] that greedy rejection sampling [45] runs in $O(\sup_z (\mathrm{d}P/\mathrm{d}Q)(z))$ time. The PPR algorithm has an additional log term due to the use of heap.

[14]A counter-based PRNG [70] allows us to directly jump to the state after $k$ uses of the PRNG, without the need of generating all $k$ samples, greatly improving the decoding efficiency. This technique is applicable to greedy rejection sampling [45] and the original Poisson functional representation [61, 60] as well.

**Algorithm 1** Poisson private representation

---

**Procedure** PPRENCODE($\alpha, Q, r, r^*, s$) :
   **Input:** parameter $\alpha > 1$, distribution $Q$, density $r(z) := (\mathrm{d}P/\mathrm{d}Q)(z)$,
        bound $r^* \geq \sup_z r(z)$, random seed $s$
   **Output:** index $k \in \mathbb{Z}_{>0}$

1: Initialize PRNG $\mathscr{G}$ using the seed $s$
2: $u \leftarrow 0$, $w^* \leftarrow \infty$, $k \leftarrow 0$, $k^* \leftarrow 0$, $n \leftarrow 0$
3: $\gamma_1 \leftarrow \gamma(1 - \alpha^{-1}, 1) = \int_0^1 e^{-\tau} \tau^{-\alpha^{-1}} \mathrm{d}\tau$
4: $h \leftarrow \emptyset$ (empty heap)
5: **while** true **do**
6:    $u \leftarrow u + \mathrm{Exp}_{\mathrm{local}}(1)$             ▷ *Generated using local randomness (not $\mathscr{G}$)*
7:    $b \leftarrow (u\alpha/(e^{-1} + \gamma_1))^\alpha$
8:    **if** $n = 0$ and $b(r^*)^{-\alpha} \geq w^*$ **then**    ▷ *No possible points left and future points impossible*
9:       **return** $k^*$
10:    **end if**
11:    **if** $\mathrm{Unif}_{\mathrm{local}}(0, 1) < e^{-1}/(e^{-1} + \gamma_1)$ **then**       ▷ *Run with prob. $e^{-1}/(e^{-1} + \gamma_1)$*
12:       $t \leftarrow b^{1/\alpha}$, $v \leftarrow \mathrm{Exp}_{\mathrm{local}}(1) + 1$
13:    **else**
14:       **repeat**
15:          $v \leftarrow \mathrm{Gamma}_{\mathrm{local}}(1 - \alpha^{-1}, 1)$        ▷ *Gamma distribution*
16:       **until** $v \leq 1$
17:       $t \leftarrow (b/v)^{1/\alpha}$
18:    **end if**
19:    $\theta \leftarrow \mathbf{1}\{(t/r^*)^\alpha v \leq w^*\}$       ▷ *Is it possible for this point to be optimal*
20:    Push $(t, v, \theta)$ to $h$
21:    $n \leftarrow n + \theta$           ▷ *Number of possible points in heap*
22:    **while** $h \neq \emptyset$ and $\min_{(t',v',\theta') \in h} t' \leq b^{1/\alpha}$ **do**   ▷ *Assign $Z_i$'s to points in heap with small $T_i$*
23:       $(t, v, \theta) \leftarrow \arg\min_{(t',v',\theta') \in h} t'$, and pop $(t, v, \theta)$ from $h$
24:       $n \leftarrow n - \theta$
25:       $k \leftarrow k + 1$
26:       Generate $z \sim Q$ using $\mathscr{G}$
27:       $w \leftarrow (t/r(z))^\alpha v$
28:       **if** $w < w^*$ **then**
29:          $w^* \leftarrow w$
30:          $k^* \leftarrow k$
31:       **end if**
32:    **end while**
33: **end while**

**Procedure** PPRDECODE($Q, k, s$) :
   **Input:** $Q$, index $k \in \mathbb{Z}_{>0}$, seed $s$
   **Output:** sample $z$

1: Initialize PRNG $\mathscr{G}$ using the seed $s$
2: **for** $i = 1, 2, \ldots, k$ **do**
3:    Generate $z \sim Q$ using $\mathscr{G}$       ▷ *See footnote 14*
4: **end for**
5: **return** $z$

for $Q$-almost every $z$,[15] and hence $e^{-\varepsilon}\tilde{T}_{2,i} \leq \tilde{T}_{1,i} \leq e^{\varepsilon}\tilde{T}_{2,i}$. For $k \in \mathbb{Z}_+$, we have, almost surely,

$$\Pr(K_1 = k \,|\, (Z_i, T_i)_i) = \frac{\tilde{T}_{1,k}^{-\alpha}}{\sum_{i=1}^{\infty} \tilde{T}_{1,i}^{-\alpha}}$$

$$\leq \frac{e^{\alpha\varepsilon}\tilde{T}_{2,k}^{-\alpha}}{\sum_{i=1}^{\infty} e^{-\alpha\varepsilon}\tilde{T}_{2,i}^{-\alpha}}$$

$$= e^{2\alpha\varepsilon} \Pr(K_2 = k \,|\, (Z_i, T_i)_i).$$

For any measurable $\mathcal{S} \subseteq \mathcal{Z}^\infty \times \mathbb{Z}_{>0}$,

$$\Pr\left(((Z_i)_i, K_1) \in \mathcal{S}\right)$$

$$= \mathbb{E}\left[\Pr\left(((Z_i)_i, K_1) \in \mathcal{S} \,\big|\, (Z_i, T_i)_i\right)\right]$$

$$= \mathbb{E}\left[\sum_{k:\,((Z_i)_i, k) \in \mathcal{S}} \Pr\left(K_1 = k \,\big|\, (Z_i, T_i)_i\right)\right]$$

$$\leq e^{2\alpha\varepsilon} \cdot \mathbb{E}\left[\sum_{k:\,((Z_i)_i, k) \in \mathcal{S}} \Pr\left(K_2 = k \,\big|\, (Z_i, T_i)_i\right)\right]$$

$$= e^{2\alpha\varepsilon} \Pr\left(((Z_i)_i, K_2) \in \mathcal{S}\right). \tag{3}$$

Hence, $P_{(Z_i)_i, K|X}$ is $2\alpha\varepsilon$-DP.

For Theorem 4.7, consider a $\varepsilon \cdot d_{\mathcal{X}}$-private mechanism $P_{Z|X}$, and consider $x_1, x_2 \in \mathcal{X}$. We have

$$e^{-\varepsilon \cdot d_{\mathcal{X}}(x_1, x_2)}\frac{\mathrm{d}P_2}{\mathrm{d}Q}(z) \leq \frac{\mathrm{d}P_1}{\mathrm{d}Q}(z) \leq e^{\varepsilon \cdot d_{\mathcal{X}}(x_1, x_2)}\frac{\mathrm{d}P_2}{\mathrm{d}Q}(z) \tag{4}$$

for $Q$-almost every $z$. By exactly the same arguments as in the proof of Theorem 4.5, $\Pr\left(((Z_i)_i, K_1) \in \mathcal{S}\right) \leq e^{2\alpha\varepsilon \cdot d_{\mathcal{X}}(x_1, x_2)} \Pr\left(((Z_i)_i, K_2) \in \mathcal{S}\right)$, and hence $P_{(Z_i)_i, K|X}$ is $2\alpha\varepsilon \cdot d_{\mathcal{X}}$-private.

## D  Proof of Theorem 4.6

Consider a $(\varepsilon, \delta)$-DP mechanism $P_{Z|X}$. Consider neighbors $x_1, x_2$, and let $P_j := P_{Z|X}(\cdot|x_j)$, and $K_j$ be the output of PPR applied on $P_j$, for $j = 1, 2$. By the definition of $(\varepsilon, \delta)$-differential privacy, we have

$$\int \max\left\{\rho_1(z) - e^\varepsilon \rho_2(z),\, 0\right\} Q(\mathrm{d}z) \leq \delta, \tag{5}$$

$$\int \max\left\{\rho_2(z) - e^\varepsilon \rho_1(z),\, 0\right\} Q(\mathrm{d}z) \leq \delta. \tag{6}$$

Let

$$\overline{\rho}(z) := \min\left\{\max\left\{\rho_1(z),\, e^{-\varepsilon}\rho_2(z)\right\},\, e^\varepsilon \rho_2(z)\right\}.$$

Note that $e^{-\varepsilon}\rho_2(z) \leq \overline{\rho}(z) \leq e^\varepsilon \rho_2(z)$. We then consider two cases:

Case 1: $\int \overline{\rho}(z)Q(\mathrm{d}z) \leq 1$. Let $\rho_3(z)$ be such that $\int \rho_3(z)Q(\mathrm{d}z) = 1$ and

$$\overline{\rho}(z) \leq \rho_3(z) \leq e^\varepsilon \rho_2(z).$$

We can always find such $\rho_3$ by taking an appropriate convex combination of the lower bound above (which integrates to $\leq 1$) and the upper obund above (which integrates to $\geq 1$). We then have

$$e^{-\varepsilon}\rho_2(z) \leq \rho_3(z) \leq e^\varepsilon \rho_2(z). \tag{7}$$

---

[15] $\varepsilon$-DP only implies that (2) holds for $P_1$-almost every $z$ (or equivalently $P_2$-almost every $z$ since $P_1, P_2$ are absolutely continuous with respect to each other). We now show that (2) holds for $Q$-almost every $z$. Apply Lebesgue's decomposition theorem to find measures $\tilde{Q}, \hat{Q}$ such that $Q = \tilde{Q} + \hat{Q}$, $\tilde{Q} \ll P_1$ and $\hat{Q} \perp P_1$. There exists $\mathcal{Z}' \subseteq \mathcal{Z}$ such that $P_1(\mathcal{Z}') = 1$ and $\hat{Q}(\mathcal{Z}') = 0$. Since $P_1 \ll Q$, we have $P_1 \ll \tilde{Q}$. We have (2) for $\tilde{Q}$-almost every $z$. Also, we have (2) for $\hat{Q}$-almost every $z$ since $z \notin \mathcal{Z}'$ gives $\frac{\mathrm{d}P_1}{\mathrm{d}Q}(z) = \frac{\mathrm{d}P_1}{\mathrm{d}\hat{Q}}(z) = 0$ for $\hat{Q}$-almost every $z$, and also $\frac{\mathrm{d}P_2}{\mathrm{d}Q}(z) = 0$ for $\hat{Q}$-almost every $z$ since $P_2 \ll P_1$.

If $\rho_1(z) - e^\varepsilon \rho_2(z) \le 0$, then $\rho_1(z) - \rho_3(z) \le \rho_1(z) - \overline{\rho}(z) \le 0$. If $\rho_1(z) - e^\varepsilon \rho_2(z) > 0$, then $\rho_3(z) = \overline{\rho}(z) = e^\varepsilon \rho_2(z)$. Either way, we have $\max\{\rho_1(z) - \rho_3(z), 0\} = \max\{\rho_1(z) - e^\varepsilon \rho_2(z), 0\}$. By (5), we have

$$\int \max\{\rho_1(z) - \rho_3(z), 0\} Q(\mathrm{d}z) \le \delta.$$

Let $P_3 = \rho_3 Q$ be the probability measure with $\mathrm{d}P_3/\mathrm{d}Q = \rho_3$. Then the total variation distance $d_{\mathrm{TV}}(P_1, P_3)$ between $P_1$ and $P_3$ is at most $\delta$, and by (7),

$$e^{-\varepsilon} \frac{\mathrm{d}P_2}{\mathrm{d}Q}(z) \le \frac{\mathrm{d}P_3}{\mathrm{d}Q}(z) \le e^{\varepsilon} \frac{\mathrm{d}P_2}{\mathrm{d}Q}(z). \tag{8}$$

Case 2: $\int \overline{\rho}(z) Q(\mathrm{d}z) > 1$. Let $\rho_3(z)$ be such that $\int \rho_3(z) Q(\mathrm{d}z) = 1$ and

$$e^{-\varepsilon} \rho_2(z) \le \rho_3(z) \le \overline{\rho}(z).$$

We can always find such $\rho_3$ by taking an appropriate convex combination of the lower bound above (which integrates to $\le 1$) and the upper obund above (which integrates to $> 1$). We again have $e^{-\varepsilon} \rho_2(z) \le \rho_3(z) \le e^{\varepsilon} \rho_2(z)$. If $e^{-\varepsilon} \rho_2(z) - \rho_1(z) \le 0$, then $\rho_3(z) - \rho_1(z) \le \overline{\rho}(z) - \rho_1(z) \le 0$. If $e^{-\varepsilon} \rho_2(z) - \rho_1(z) > 0$, then $\rho_3(z) = \overline{\rho}(z) = e^{-\varepsilon} \rho_2(z)$. Either way, we have $\max\{\rho_3(z) - \rho_1(z), 0\} = \max\{e^{-\varepsilon} \rho_2(z) - \rho_1(z), 0\}$. By (6), we have

$$\int \max\{\rho_3(z) - \rho_1(z), 0\} Q(\mathrm{d}z) \le e^{-\varepsilon} \delta \le \delta.$$

Let $P_3 = \rho_3 Q$ be the probability measure with $\mathrm{d}P_3/\mathrm{d}Q = \rho_3$. Again, we have $d_{\mathrm{TV}}(P_1, P_3) \le \delta$ and (8). Therefore, regardless of whether Case 1 or Case 2 holds, we can construct $P_3$ satisfying $d_{\mathrm{TV}}(P_1, P_3) \le \delta$ and (8). Let $K_3$ be the output of PPR applied on $P_3$.

In the proof of Theorem F.1, we see that PPR has the following equivalent formulation. Let $(T_i)_i \sim \mathrm{PP}(1)$ be a Poisson process with rate 1, independent of $Z_1, Z_2, \ldots \overset{iid}{\sim} Q$. Let $R_i := (\mathrm{d}P/\mathrm{d}Q)(Z_i)$, and let its probability measure be $P_R$. Let $V_1, V_2, \ldots \overset{iid}{\sim} \mathrm{Exp}(1)$. PPR can be equivalently expressed as

$$K = \underset{k}{\mathrm{argmin}}\, T_k^\alpha R_k^{-\alpha} V_k = \underset{k}{\mathrm{argmin}}\, \frac{T_k V_k^{1/\alpha}}{R_k}.$$

Note that $(T_i V_i^{1/\alpha})_i \sim \mathrm{PP}(\int_0^\infty v^{-1/\alpha} e^{-v} \mathrm{d}v) = \mathrm{PP}(\Gamma(1 - \alpha^{-1}))$ is a uniform Poisson process. Therefore PPR is the same as the Poisson functional representation [61, 60] applied on $(T_i V_i^{1/\alpha})_i$. By the grand coupling property of Poisson functional representation [60, 59] (see [59, Theorem 3]), if we apply the Poisson functional representation on $P_1$ and $P_3$ to get $K_1$ and $K_3$ respectively, then

$$\Pr(K_1 \ne K_3) \le 2d_{\mathrm{TV}}(P_1, P_3) \le 2\delta.$$

Therefore, for any measurable $\mathcal{S} \subseteq \mathcal{Z}^\infty \times \mathbb{Z}_{>0}$,

$$\Pr(((Z_i)_i, K_1) \in \mathcal{S}) \le \Pr(((Z_i)_i, K_3) \in \mathcal{S}) + 2\delta$$
$$\le e^{2\alpha\varepsilon} \Pr(((Z_i)_i, K_2) \in \mathcal{S}) + 2\delta,$$

where the last inequality is by applying (3) on $P_3, P_2$ instead of $P_1, P_2$. Hence, $P_{(Z_i)_i, K|X}$ is $(2\alpha\varepsilon, 2\delta)$-DP.

## E  Proof of Theorem 4.8

We present the proof of $(\varepsilon, \delta)$-DP of PPR (i.e., Theorem 4.8).

*Proof.* We assume

$$\alpha - 1 \le \frac{\beta \tilde{\delta} \tilde{\varepsilon}^2}{-\ln \tilde{\delta}}, \tag{9}$$

where $\beta := e^{-4.2}$. Using the Laplace functional of the Poisson process $(\tilde{T}_i)_i$ [57, Theorem 3.9], for $w > 0$,

$$\mathbb{E}\left[\exp\left(-w\sum_i \tilde{T}_i^{-\alpha}\right)\right] = \exp\left(-\int_0^\infty (1 - \exp(-wt^{-\alpha}))\mathrm{d}t\right) \tag{10}$$

$$= \exp\left(-w^{1/\alpha}\Gamma(1 - \alpha^{-1})\right).$$

We first bound the left tail of $\sum_i \tilde{T}_i^{-\alpha}$. By Chernoff bound, for $d \geq 0$,

$$\Pr\left(\sum_i \tilde{T}_i^{-\alpha} \leq d\right)$$

$$\leq \inf_{w>0} e^{wd}\mathbb{E}\left[\exp\left(-w\sum_i \tilde{T}_i^{-\alpha}\right)\right]$$

$$= \inf_{w>0} \exp\left(wd - w^{1/\alpha}\Gamma(1 - \alpha^{-1})\right)$$

$$\leq \exp\left(\left(\frac{\Gamma(1 - \alpha^{-1})}{\alpha d}\right)^{\frac{\alpha}{\alpha-1}} d - \left(\frac{\Gamma(1 - \alpha^{-1})}{\alpha d}\right)^{\frac{1}{\alpha-1}} \Gamma(1 - \alpha^{-1})\right)$$

$$= \exp\left((\Gamma(1 - \alpha^{-1}))^{\frac{\alpha}{\alpha-1}} d^{-\frac{1}{\alpha-1}}\left(\alpha^{-\frac{\alpha}{\alpha-1}} - \alpha^{-\frac{1}{\alpha-1}}\right)\right)$$

$$= \exp\left(-\left(\frac{\alpha d}{(\Gamma(1 - \alpha^{-1}))^\alpha}\right)^{-\frac{1}{\alpha-1}}\left(1 - \alpha^{-1}\right)\right)$$

$$= \exp\left(-\left(\frac{\alpha d(1 - \alpha^{-1})^\alpha}{(\Gamma(2 - \alpha^{-1}))^\alpha}\right)^{-\frac{1}{\alpha-1}}\left(1 - \alpha^{-1}\right)\right)$$

$$= \exp\left(-\left(\frac{(\alpha - 1)d}{(\Gamma(2 - \alpha^{-1}))^\alpha}\right)^{-\frac{1}{\alpha-1}}\right).$$

Therefore, to guarantee $\Pr(\sum_i \tilde{T}_i^{-\alpha} \leq d) \leq \tilde{\delta}/3$, we require

$$d \leq \frac{\Gamma(2 - \alpha^{-1})^\alpha\left(-\ln(\tilde{\delta}/3)\right)^{-(\alpha-1)}}{\alpha - 1},$$

where

$$\Gamma(2 - \alpha^{-1})^\alpha\left(-\ln(\tilde{\delta}/3)\right)^{-(\alpha-1)}$$

$$\geq (\exp(-\gamma(\alpha - 1)))^\alpha\left(-\ln(\tilde{\delta}^2)\right)^{-(\alpha-1)}$$

$$\geq \exp\left(-\gamma\alpha\frac{\beta\tilde{\delta}\tilde{\varepsilon}^2}{-\ln\tilde{\delta}}\right)\left(-2\ln\tilde{\delta}\right)^{-\frac{\beta\tilde{\delta}\tilde{\varepsilon}^2}{-\ln\tilde{\delta}}}$$

$$\geq \exp\left(-2\gamma\frac{\beta\tilde{\delta}\tilde{\varepsilon}^2}{-\ln\tilde{\delta}} - 2e^{-1}\beta\tilde{\delta}\tilde{\varepsilon}^2\right)$$

$$\geq \exp\left(-\left(\frac{2\gamma}{3\ln 2} + \frac{2}{3e}\right)\beta\tilde{\varepsilon}^2\right)$$

$$\geq \exp\left(-0.81 \cdot \beta\tilde{\varepsilon}^2\right)$$

$$\geq e^{-\tilde{\varepsilon}/2},$$

since $1 < \alpha \leq 2$, $0 < \tilde{\delta} \leq 1/3$, $\beta = e^{-4.2}$ and $0 < \tilde{\varepsilon} \leq 1$, where $\gamma$ is the Euler-Mascheroni constant. Hence, we have

$$\Pr\left(\sum_i \tilde{T}_i^{-\alpha} \leq \frac{e^{-\tilde{\varepsilon}/2}}{\alpha - 1}\right) \leq \frac{\tilde{\delta}}{3}. \tag{11}$$

We then bound the right tail of $\sum_i \tilde{T}_i^{-\alpha}$. Unfortunately, the Laplace functional (10) does not work since the integral diverges for small $t$. Therefore, we have to bound $t$ away from $0$. Note that $\min_i \tilde{T}_i \sim \text{Exp}(1)$, and hence

$$\Pr(\min_i \tilde{T}_i \leq \tilde{\delta}/3) \leq \tilde{\delta}/3. \tag{12}$$

Write $\tau = \tilde{\delta}/3$. Using the Laplace functional again, for $w > 0$,

$$\mathbb{E}\left[\exp\left(w \sum_{i:\tilde{T}_i > \tau} \tilde{T}_i^{-\alpha}\right)\right]$$
$$= \exp\left(-\int_\tau^\infty (1 - \exp(wt^{-\alpha}))\mathrm{d}t\right)$$
$$= \exp\left(\int_\tau^\infty (\exp(wt^{-\alpha}) - 1)\mathrm{d}t\right)$$
$$\leq \exp\left(\int_\tau^\infty (\exp(w\tau^{-\alpha}) - 1)\frac{t^{-\alpha}}{\tau^{-\alpha}}\mathrm{d}t\right)$$
$$= \exp\left(\frac{\exp(w\tau^{-\alpha}) - 1}{\tau^{-\alpha}} \cdot \frac{\tau^{-(\alpha-1)}}{\alpha - 1}\right)$$
$$= \exp\left(\frac{\tau(\exp(w\tau^{-\alpha}) - 1)}{\alpha - 1}\right).$$

Therefore, by Chernoff bound, for $d \geq 0$,

$$\Pr\left(\sum_{i:\tilde{T}_i > \tau} \tilde{T}_i^{-\alpha} \geq d\right)$$
$$\leq \inf_{w>0} \exp\left(-wd + \frac{\tau(\exp(w\tau^{-\alpha}) - 1)}{\alpha - 1}\right)$$
$$\leq \exp\left(-d\tau^\alpha \ln(d(\alpha - 1)\tau^{\alpha-1}) + \frac{\tau(\exp(\ln(d(\alpha-1)\tau^{\alpha-1})) - 1)}{\alpha - 1}\right)$$
$$= \exp\left(-d\tau^\alpha \ln(d(\alpha - 1)\tau^{\alpha-1}) + \tau\frac{d(\alpha - 1)\tau^{\alpha-1} - 1}{\alpha - 1}\right)$$
$$= \exp\left(-\frac{c\tau}{\alpha - 1} \ln c + \tau\frac{c - 1}{\alpha - 1}\right)$$
$$= \exp\left(-\frac{\tau}{\alpha - 1}(c \ln c - c + 1)\right)$$
$$\leq \exp\left(-\frac{\tau(2 \ln 2 - 1)(c - 1)^2}{\alpha - 1}\right), \tag{13}$$

where

$$c := d(\alpha - 1)\tau^{\alpha-1},$$

and the last inequality holds whenever $c \in [1, 2]$ since in this range,

$$c \ln c - c + 1 \geq (2 \ln 2 - 1)(c - 1)^2.$$

Substituting

$$d = \frac{e^{\tilde{\varepsilon}/2}}{\alpha - 1},$$

we have $c = e^{\tilde{\varepsilon}/2}\tau^{\alpha-1}$. By (13), to guarantee $\Pr(\sum_{i:\tilde{T}_i > \tau} \tilde{T}_i^{-\alpha} \geq d) \leq \tilde{\delta}/3 = \tau$, we require

$$\frac{\tau(2 \ln 2 - 1)(e^{\tilde{\varepsilon}/2}\tau^{\alpha-1} - 1)^2}{\alpha - 1} \geq -\ln \tau,$$

$$e^{\tilde{\varepsilon}/2}\tau^{\alpha-1} \geq \sqrt{\frac{(\alpha-1)(-\ln\tau)}{\tau(2\ln 2-1)}+1}. \tag{14}$$

Substituting (9), we have

$$e^{\tilde{\varepsilon}/2}\tau^{\alpha-1} \geq e^{\tilde{\varepsilon}/2}\tau^{\frac{\beta\tilde{\delta}\tilde{\varepsilon}^2}{-\ln\tilde{\delta}}}$$

$$= \exp\left(\frac{\tilde{\varepsilon}}{2}+\left(\ln\frac{\tilde{\delta}}{3}\right)\frac{\beta\tilde{\delta}\tilde{\varepsilon}^2}{-\ln\tilde{\delta}}\right)$$

$$\geq \exp\left(\frac{\tilde{\varepsilon}}{2}+\left(2\ln\tilde{\delta}\right)\frac{\beta\tilde{\varepsilon}}{-3\ln\tilde{\delta}}\right)$$

$$= \exp\left(\tilde{\varepsilon}\left(\frac{1}{2}-\frac{2\beta}{3}\right)\right),$$

since $0 < \tilde{\delta} \leq 1/3$. Note that this also guarantees $c = e^{\tilde{\varepsilon}/2}\tau^{\alpha-1} \in [1,2]$ since $\beta = e^{-4.2}$ and $0 < \tilde{\varepsilon} \leq 1$. We also have

$$\frac{(\alpha-1)(-\ln\tau)}{\tau(2\ln 2-1)} \leq \frac{\frac{\beta\tilde{\delta}\tilde{\varepsilon}^2}{-\ln\tilde{\delta}}(-\ln\tau)}{\tau(2\ln 2-1)}$$

$$\leq \frac{\frac{\beta\tilde{\delta}\tilde{\varepsilon}^2}{-\ln\tilde{\delta}}(-2\ln\tilde{\delta})}{(\tilde{\delta}/3)(2\ln 2-1)}$$

$$= \frac{6\beta\tilde{\varepsilon}^2}{2\ln 2-1}$$

$$\leq 16\beta\tilde{\varepsilon}^2.$$

Hence,

$$\sqrt{\frac{(\alpha-1)(-\ln\tau)}{\tau(2\ln 2-1)}+1} \leq 4\tilde{\varepsilon}\sqrt{\beta}+1$$

$$\leq \exp\left(4\tilde{\varepsilon}\sqrt{\beta}\right)$$

$$\overset{(a)}{\leq} \exp\left(\tilde{\varepsilon}\left(\frac{1}{2}-\frac{2\beta}{3}\right)\right)$$

$$\leq e^{\tilde{\varepsilon}/2}\tau^{\alpha-1},$$

where (a) is by $\beta = e^{-4.2}$. Hence (14) is satisfied, and

$$\Pr\left(\sum_{i:\tilde{T}_i>\tau}\tilde{T}_i^{-\alpha} \geq \frac{e^{\tilde{\varepsilon}/2}}{\alpha-1}\right) \leq \frac{\tilde{\delta}}{3}.$$

Combining this with (11) and (12),

$$\Pr\left(\sum_i \tilde{T}_i^{-\alpha} \notin \left[\frac{e^{-\tilde{\varepsilon}/2}}{\alpha-1},\frac{e^{\tilde{\varepsilon}/2}}{\alpha-1}\right]\right)$$

$$\leq \Pr\left(\sum_i \tilde{T}_i^{-\alpha} \leq \frac{e^{-\tilde{\varepsilon}/2}}{\alpha-1}\right) + \Pr(\min_i \tilde{T}_i \leq \tilde{\delta}/3)$$

$$+ \Pr\left(\sum_{i:\tilde{T}_i>\tilde{\delta}/3}\tilde{T}_i^{-\alpha} \geq \frac{e^{\tilde{\varepsilon}/2}}{\alpha-1}\right)$$

$$\leq \tilde{\delta}.$$

Consider an $(\varepsilon,\delta)$-differentially private mechanism $P_{Z|X}$. Consider neighbors $x_1, x_2$, and let $P_j := P_{Z|X}(\cdot|x_j)$, $\tilde{T}_{j,i} := T_i/(\frac{\mathrm{d}P_j}{\mathrm{d}Q}(Z_i))$, and $K_j$ be the output of PPR applied on $P_j$, for $j = 1, 2$. We

first consider the case $\delta = 0$, which gives $\frac{\mathrm{d}P_1}{\mathrm{d}Q}(z) \le e^\varepsilon \frac{\mathrm{d}P_2}{\mathrm{d}Q}(z)$ for every $z$. For any measurable $\mathcal{S} \subseteq \mathcal{Z}^\infty \times \mathbb{Z}_{>0}$,

$$
\begin{aligned}
&\Pr\left(((Z_i)_i, K_1) \in \mathcal{S}\right) \\
&= \mathbb{E}\left[\Pr\left(((Z_i)_i, K_1) \in \mathcal{S} \,\big|\, (Z_i, T_i)_i\right)\right] \\
&= \mathbb{E}\left[\sum_{k:\,((Z_i)_i, k) \in \mathcal{S}} \Pr\left(K_1 = k \,\big|\, (Z_i, T_i)_i\right)\right] \\
&= \mathbb{E}\left[\sum_{k:\,((Z_i)_i, k) \in \mathcal{S}} \frac{\tilde{T}_{1,k}^{-\alpha}}{\sum_i \tilde{T}_{1,i}^{-\alpha}}\right] \\
&\le \mathbb{E}\left[\mathbf{1}\left\{\sum_i \tilde{T}_{1,i}^{-\alpha} \in \left[\frac{e^{-\tilde\varepsilon/2}}{\alpha-1}, \frac{e^{\tilde\varepsilon/2}}{\alpha-1}\right]\right\} \min\left\{\sum_{k:\,((Z_i)_i, k) \in \mathcal{S}} \frac{\tilde{T}_{1,k}^{-\alpha}}{\sum_i \tilde{T}_{1,i}^{-\alpha}}, 1\right\}\right] + \tilde\delta \\
&\le \mathbb{E}\left[\min\left\{\sum_{k:\,((Z_i)_i, k) \in \mathcal{S}} \frac{\tilde{T}_{1,k}^{-\alpha}}{e^{-\tilde\varepsilon/2}/(\alpha-1)}, 1\right\}\right] + \tilde\delta \\
&= \mathbb{E}\left[\min\left\{\sum_{k:\,((Z_i)_i, k) \in \mathcal{S}} \frac{(\frac{\mathrm{d}P_1}{\mathrm{d}Q}(Z_k))^\alpha T_k^{-\alpha}}{e^{-\tilde\varepsilon/2}/(\alpha-1)}, 1\right\}\right] + \tilde\delta \\
&\le \mathbb{E}\left[\min\left\{\sum_{k:\,((Z_i)_i, k) \in \mathcal{S}} \frac{(e^\varepsilon \frac{\mathrm{d}P_2}{\mathrm{d}Q}(Z_k))^\alpha T_k^{-\alpha}}{e^{-\tilde\varepsilon/2}/(\alpha-1)}, 1\right\}\right] + \tilde\delta \\
&\le \mathbb{E}\left[\mathbf{1}\left\{\sum_i \tilde{T}_{2,i}^{-\alpha} \in \left[\frac{e^{-\tilde\varepsilon/2}}{\alpha-1}, \frac{e^{\tilde\varepsilon/2}}{\alpha-1}\right]\right\} \min\left\{\sum_{k:\,((Z_i)_i, k) \in \mathcal{S}} \frac{(e^\varepsilon \frac{\mathrm{d}P_2}{\mathrm{d}Q}(Z_k))^\alpha T_k^{-\alpha}}{e^{-\tilde\varepsilon/2}/(\alpha-1)}, 1\right\}\right] + 2\tilde\delta \\
&\le \mathbb{E}\left[\min\left\{e^{\alpha\varepsilon} \sum_{k:\,((Z_i)_i, k) \in \mathcal{S}} \frac{(\frac{\mathrm{d}P_2}{\mathrm{d}Q}(Z_k))^\alpha T_k^{-\alpha}}{e^{-\tilde\varepsilon} \sum_i \tilde{T}_{2,i}^{-\alpha}}, 1\right\}\right] + 2\tilde\delta \\
&\le \mathbb{E}\left[e^{\alpha\varepsilon + \tilde\varepsilon} \sum_{k:\,((Z_i)_i, k) \in \mathcal{S}} \frac{\tilde{T}_{2,k}^{-\alpha}}{\sum_i \tilde{T}_{2,i}^{-\alpha}}\right] + 2\tilde\delta \\
&= e^{\alpha\varepsilon + \tilde\varepsilon} \Pr\left(((Z_i)_i, K_2) \in \mathcal{S}\right) + 2\tilde\delta.
\end{aligned}
\tag{15}
$$

Hence PPR is $(\alpha\varepsilon + \tilde\varepsilon, 2\tilde\delta)$-differentially private.

For the case $\delta > 0$, by the definition of $(\varepsilon, \delta)$-differential privacy, we have

$$
\int \max\left\{\frac{\mathrm{d}P_1}{\mathrm{d}Q}(z) - e^\varepsilon \frac{\mathrm{d}P_2}{\mathrm{d}Q}(z), 0\right\} Q(\mathrm{d}z) \le \delta.
$$

Let $P_3$ be a probability measure that satisfies

$$
\min\left\{\frac{\mathrm{d}P_1}{\mathrm{d}Q}(z), e^\varepsilon \frac{\mathrm{d}P_2}{\mathrm{d}Q}(z)\right\} \le \frac{\mathrm{d}P_3}{\mathrm{d}Q}(z) \le e^\varepsilon \frac{\mathrm{d}P_2}{\mathrm{d}Q}(z),
$$

for every $z$. Such $P_3$ can be constructed by taking an appropriate convex combination of the lower bound above (which integrates to $\le 1$) and the upper bound above (which integrates to $\ge 1$) such that $P_3$ integrates to 1. We have

$$
\int \max\left\{\frac{\mathrm{d}P_1}{\mathrm{d}Q}(z) - \frac{\mathrm{d}P_3}{\mathrm{d}Q}(z), 0\right\} Q(\mathrm{d}z) \le \delta,
$$

and hence the total variation distance $d_{\mathrm{TV}}(P_1, P_3)$ between $P_1$ and $P_3$ is at most $\delta$. Let $K_3$ be the output of PPR applied on $P_3$.

In the proof of Theorem F.1, we see that PPR has the following equivalent formulation. Let $(T_i)_i \sim$ PP(1) be a Poisson process with rate 1, independent of $Z_1, Z_2, \ldots \overset{iid}{\sim} Q$. Let $R_i := (\mathrm{d}P/\mathrm{d}Q)(Z_i)$, and let its probability measure be $P_R$. Let $V_1, V_2, \ldots \overset{iid}{\sim} \mathrm{Exp}(1)$. PPR can be equivalently expressed as

$$K = \underset{k}{\operatorname{argmin}} T_k^\alpha R_k^{-\alpha} V_k = \underset{k}{\operatorname{argmin}} \frac{T_k V_k^{1/\alpha}}{R_k}.$$

Note that $(T_i V_i^{1/\alpha})_i \sim \mathrm{PP}(\int_0^\infty v^{-1/\alpha} e^{-v} \mathrm{d}v) = \mathrm{PP}(\Gamma(1 - \alpha^{-1}))$ is a uniform Poisson process. Therefore PPR is the same as the Poisson functional representation [61, 60] applied on $(T_i V_i^{1/\alpha})_i$. By the grand coupling property of Poisson functional representation [60, 59] (see [59, Theorem 3]), if we apply the Poisson functional representation on $P_1$ and $P_3$ to get $K_1$ and $K_3$ respectively, then

$$\Pr(K_1 \neq K_3) \leq 2d_{\mathrm{TV}}(P_1, P_3) \leq 2\delta.$$

Therefore, for any measurable $S \subseteq \mathcal{Z}^\infty \times \mathbb{Z}_{>0}$,

$$\begin{aligned} &\Pr\left(((Z_i)_i, K_1) \in S\right) \\ &\leq \Pr\left(((Z_i)_i, K_3) \in S\right) + 2\delta \\ &\leq e^{\alpha\varepsilon + \tilde{\varepsilon}} \Pr\left(((Z_i)_i, K_2) \in S\right) + 2\tilde{\delta} + 2\delta, \end{aligned}$$

where the last inequality is by applying (15) on $P_3, P_2$ instead of $P_1, P_2$. This completes the proof.

$\square$

# F   Proof of Theorem 4.3

We now bound the size of the index output by the Poisson private representation. The following is a refined version of Theorem 4.3.

**Theorem F.1.** *For PPR with parameter $\alpha > 1$, when the encoder is given the input $x$, the message $K$ given by PPR satisfies*

$$\mathbb{E}[\log K] \leq D(P\|Q)$$

$$+ \inf_{\eta \in (0,1] \cap (0, \alpha-1)} \frac{1}{\eta} \log\left(\frac{\Gamma(1 - \frac{\eta+1}{\alpha})\Gamma(\eta+1)}{(\Gamma(1 - \frac{1}{\alpha}))^{\eta+1}} + 1\right) \tag{16}$$

$$\leq D(P\|Q) + \frac{\log(3.56)}{\min\{(\alpha-1)/2, 1\}}, \tag{17}$$

*where $P := P_{Z|X}(\cdot|x)$.*

Note that for $\alpha = \infty$, (16) with $\eta = 1$ gives $\mathbb{E}[\log K] \leq D(P\|Q) + \log 2$, recovering the bound in [58] (which strengthened [61]).

*Proof.* Write $(X_i)_i \sim \mathrm{PP}(\mu)$ if the points $(X_i)_i$ (as a multiset, ignoring the ordering) form a Poisson point process with intensity measure $\mu$. Similarly, for $f : [0, \infty)^n \to [0, \infty)$, we write $\mathrm{PP}(f)$ for the Poisson point process with intensity function $f$ (i.e., the intensity measure has a Radon-Nikodym derivative $f$ against the Lebesgue measure). Let $(T_i)_i \sim \mathrm{PP}(1)$ be a Poisson process with rate 1, independent of $Z_1, Z_2, \ldots \overset{iid}{\sim} Q$. Let $R_i := (\mathrm{d}P/\mathrm{d}Q)(Z_i)$, and let its probability measure be $P_R$. We have $\tilde{T}_i = T_i/R_i$. Let $V_1, V_2, \ldots \overset{iid}{\sim} \mathrm{Exp}(1)$. By the property of exponential random variables, for any $p_1, p_2, \ldots \geq 0$ with $\sum_i p_i < \infty$, we have $\Pr(\operatorname{argmin}_k V_k/p_k = k) = p_k / \sum_i p_i$. Therefore, PPRF can be equivalently expressed as

$$K = \underset{k}{\operatorname{argmin}} T_k^\alpha R_k^{-\alpha} V_k.$$

By the marking theorem [57], $(T_i, R_i, V_i)_i$ is a Poisson process over $[0, \infty)^3$ with intensity measure

$$(T_i, R_i, V_i)_i \sim \mathrm{PP}\left(e^{-v} P_R(r)\right).$$

By the mapping theorem [57], letting $W_i := T_i^\alpha R_i^{-\alpha} V_i$, we have

$$(T_i, R_i, W_i)_i \sim \mathrm{PP}\left(r^\alpha t^{-\alpha} e^{-wr^\alpha t^{-\alpha}} P_R(r)\right). \tag{18}$$

Again by the mapping theorem,

$$
\begin{aligned}
(W_i)_i &\sim \mathrm{PP}\left(\mathbb{E}_{R\sim P_R}\left[\int_0^\infty R^\alpha t^{-\alpha} e^{-wR^\alpha t^{-\alpha}} \mathrm{d}t\right]\right) \\
&= \mathrm{PP}\left(\mathbb{E}\left[\alpha^{-1}(wR^\alpha)^{1/\alpha-1}\Gamma(1-\alpha^{-1})R^\alpha\right]\right) \\
&= \mathrm{PP}\left(\mathbb{E}\left[\alpha^{-1}w^{1/\alpha-1}\Gamma(1-\alpha^{-1})R\right]\right) \\
&= \mathrm{PP}\left(\alpha^{-1}w^{1/\alpha-1}\Gamma(1-\alpha^{-1})\right)
\end{aligned}
$$

since $\mathbb{E}[R] = \int(\mathrm{d}P/\mathrm{d}Q)(z)Q(\mathrm{d}z) = 1$. Recall that $W_K = \min_i W_i$ by the definition of $K$. We have

$$
\begin{aligned}
\mathrm{Pr}(W_K > w) &= \exp\left(-\int_0^w \alpha^{-1}v^{1/\alpha-1}\Gamma(1-\alpha^{-1})\mathrm{d}v\right) \\
&= \exp\left(-w^{1/\alpha}\Gamma(1-\alpha^{-1})\right).
\end{aligned}
$$

Hence the probability density function of $W_K$ is

$$
\begin{aligned}
&-\frac{\mathrm{d}}{\mathrm{d}w}\exp\left(-w^{1/\alpha}\Gamma(1-\alpha^{-1})\right) \\
&= \alpha^{-1}w^{1/\alpha-1}\Gamma(1-\alpha^{-1})\exp\left(-w^{1/\alpha}\Gamma(1-\alpha^{-1})\right). \tag{19}
\end{aligned}
$$

By (18), the Radon-Nikodym derivative between the conditional distribution of $R_K$ given $W_K = w$ and $P_R$ is

$$
\begin{aligned}
&\mathrm{Pr}(R_K \in [r, r+\mathrm{d}r) \,|\, W_K = w)/P_R(\mathrm{d}r) \\
&= \frac{\int_0^\infty r^\alpha t^{-\alpha} e^{-wr^\alpha t^{-\alpha}} \mathrm{d}t}{\mathbb{E}_{R\sim P_R}\left[\int_0^\infty R^\alpha t^{-\alpha} e^{-wR^\alpha t^{-\alpha}} \mathrm{d}t\right]} \\
&= \frac{\alpha^{-1}w^{1/\alpha-1}\Gamma(1-\alpha^{-1})r}{\alpha^{-1}w^{1/\alpha-1}\Gamma(1-\alpha^{-1})} \\
&= r
\end{aligned}
$$

does not depend on $w$. Hence $R_K$ is independent of $W_K$. By (18), for $0 \le \eta < \alpha - 1$,

$$
\begin{aligned}
&\mathbb{E}[T_K^\eta \,|\, R_K = r, W_K = w] \\
&= \frac{\int_0^\infty t^\eta r^\alpha t^{-\alpha} e^{-wr^\alpha t^{-\alpha}} \mathrm{d}t}{\int_0^\infty r^\alpha t^{-\alpha} e^{-wr^\alpha t^{-\alpha}} \mathrm{d}t} \\
&= \frac{\alpha^{-1}w^{(\eta+1)/\alpha-1}\Gamma(1-(\eta+1)\alpha^{-1})r^{\eta+1}}{\alpha^{-1}w^{1/\alpha-1}\Gamma(1-\alpha^{-1})r}. \tag{20}
\end{aligned}
$$

Since $R_K$ is independent of $W_K$, using (20) and (19), for $\eta \in (0, 1] \cap (0, \alpha - 1)$,

$$
\begin{aligned}
&\mathbb{E}[T_K^\eta \,|\, R_K = r] \\
&= \int_0^\infty \alpha^{-1}w^{(\eta+1)/\alpha-1}\Gamma(1-(\eta+1)\alpha^{-1})r^\eta \exp\left(-w^{1/\alpha}\Gamma(1-\alpha^{-1})\right)\mathrm{d}w \\
&= r^\eta\Gamma(1-(\eta+1)\alpha^{-1})\int_0^\infty \alpha^{-1}w^{(\eta+1)/\alpha-1}\exp\left(-w^{1/\alpha}\Gamma(1-\alpha^{-1})\right)\mathrm{d}w \\
&= r^\eta\Gamma(1-(\eta+1)\alpha^{-1})(\Gamma(1-\alpha^{-1}))^{-(\eta+1)}\Gamma(\eta+1) \\
&=: c_{\alpha,\eta}r^\eta, \tag{21}
\end{aligned}
$$

where $c_{\alpha,\eta} := \Gamma(1 - (\eta + 1)\alpha^{-1})(\Gamma(1 - \alpha^{-1}))^{-(\eta+1)}\Gamma(\eta + 1)$. Hence,

$$\mathbb{E}[\log(T_K + 1) \mid R_K = r]$$
$$\leq \mathbb{E}[\log((T_K^\eta + 1)^{1/\eta}) \mid R_K = r]$$
$$= \mathbb{E}[\eta^{-1} \log(T_K^\eta + 1) \mid R_K = r]$$
$$\leq \eta^{-1} \log(c_{\alpha,\eta} r^\eta + 1). \tag{22}$$

Note that
$$K - 1 = |\{i : T_i < T_K\}|,$$
and hence the expecation of $K-1$ given $T_K$ should be around $T_K$. This is not exact since conditioning on $T_K$ changes the distribution of the process $(T_i, R_i, V_i)_i$. To resolve this problem, we define a new process $(T_i', R_i', V_i')_i$ which includes all points in $(T_i, R_i, V_i)_i$ excluding the point $(T_K, R_K, V_K)$, together with newly generated points according to
$$\mathrm{PP}\left(e^{-v} P_R(r) \mathbf{1}\{t^\alpha r^{-\alpha} v < T_K^\alpha R_K^{-\alpha} V_K\}\right).$$

Basically, $\{t^\alpha r^{-\alpha} v < T_K^\alpha R_K^{-\alpha} V_K\}$ is the "impossible region" where the points in $(T_i, R_i, V_i)_i$ cannot be located in, since $K$ attains the minimum of $T_K^\alpha R_K^{-\alpha} V_K$. The new process $(T_i', R_i', V_i')_i$ removes the point $(T_K, R_K, V_K)$, and then fills in the impossible region. It is straightforward to check that $(T_i', R_i', V_i')_i \sim \mathrm{PP}(e^{-v} P_R(r))$, independent of $(T_K, R_K, V_K)$. We have

$$\mathbb{E}[K \mid T_K]$$
$$= \mathbb{E}\left[|\{i : T_i < T_K\}| \,\Big|\, T_K\right] + 1$$
$$\leq \mathbb{E}\left[|\{i : T_i' < T_K\}| \,\Big|\, T_K\right] + 1$$
$$= T_K + 1.$$

Therefore, by (22) and Jensen's inequality,

$$\mathbb{E}[\log K]$$
$$= \mathbb{E}\left[\mathbb{E}[\log K \mid T_K]\right]$$
$$\leq \mathbb{E}\left[\log(T_K + 1)\right]$$
$$= \mathbb{E}\left[\mathbb{E}\left[\log(T_K + 1) \mid R_K\right]\right]$$
$$\leq \mathbb{E}\left[\eta^{-1} \log(c_{\alpha,\eta} R_K^\eta + 1)\right]$$
$$= \eta^{-1} \mathbb{E}_{Z \sim P}\left[\log\left(c_{\alpha,\eta}\left(\frac{\mathrm{d}P}{\mathrm{d}Q}(Z)\right)^\eta + 1\right)\right]$$
$$= \eta^{-1} \mathbb{E}\left[\log\left(\left(\frac{\mathrm{d}P}{\mathrm{d}Q}(Z)\right)^\eta\right)\right] + \eta^{-1} \mathbb{E}\left[\log\left(c_{\alpha,\eta} + \left(\frac{\mathrm{d}P}{\mathrm{d}Q}(Z)\right)^{-\eta}\right)\right]$$
$$\leq D(P\|Q) + \eta^{-1} \log\left(c_{\alpha,\eta} + \left(\mathbb{E}\left[\left(\frac{\mathrm{d}P}{\mathrm{d}Q}(Z)\right)^{-1}\right]\right)^\eta\right)$$
$$\leq D(P\|Q) + \eta^{-1} \log(c_{\alpha,\eta} + 1),$$

where the last line is due to $\mathbb{E}[((\mathrm{d}P/\mathrm{d}Q)(Z))^{-1}] = \int ((\mathrm{d}P/\mathrm{d}Q)(Z))^{-1} P(\mathrm{d}z) \leq 1$ (this step appeared in [58]). The bound (16) follows from minimizing over $\eta \in (0, 1] \cap (0, \alpha - 1)$.

To show (17), substituting $\eta = \min\{(\alpha - 1)/2, 1\}$,

$$c_{\alpha,\eta} = \frac{\Gamma(1 - (\eta + 1)\alpha^{-1})\Gamma(\eta + 1)}{(\Gamma(1 - \alpha^{-1}))^{\eta+1}}$$
$$\overset{(a)}{\leq} \frac{(1 - \alpha^{-1})^{\eta+1}}{0.885^{\eta+1} \cdot (1 - (\eta + 1)\alpha^{-1})}$$
$$\leq \frac{(1 - \alpha^{-1})^{\eta+1}}{0.885^2 \cdot (1 - ((\alpha - 1)/2 + 1)\alpha^{-1})}$$
$$= \frac{2}{0.885^2}(1 - \alpha^{-1})^\eta$$
$$\leq 2.56,$$

where (a) is because $0.885 \leq x\Gamma(x) = \Gamma(x+1) \leq 1$ for $0 < x \leq 1$. Hence,

$$\mathbb{E}[\log K] \leq D(P\|Q) + \eta^{-1}\log(c_{\alpha,\eta} + 1),$$

$$\leq D(P\|Q) + \frac{\log(3.56)}{\min\{(\alpha-1)/2, 1\}}.$$

$\square$

# G  Distributed Mean Estimation with Rényi DP

In many machine learning applications, privacy budgets are often accounted in the moment space, and one popular moment accountant is the Rényi DP accountant. For completeness, we provide a Rényi DP version of Corollary 5.2 in this section. We begin with the following definition of Rényi DP:

**Definition G.1** (Rényi Differential privacy [1, 68]). Given a mechanism $\mathcal{A}$ which induces the conditional distribution $P_{Z|X}$ of $Z = \mathcal{A}(X)$, we say that it satisfies $(\gamma, \varepsilon)$- Rényi DP if for any neighboring $(x, x') \in \mathcal{N}$ and $\mathcal{S} \subseteq \mathcal{Z}$, it holds that

$$D_\gamma\left(P_{Z|X=x}\big\|P_{Z|X=x'}\right) \leq \varepsilon,$$

where

$$D_\gamma\left(P\|Q\right) := \frac{1}{\gamma-1}\log\left(\mathbb{E}_Q\left[\left(\frac{P}{Q}\right)^\gamma\right]\right)$$

is the Rényi divergence between $P$ and $Q$. If $\mathcal{N} = \mathcal{X}^2$, we say that the mechanism satisfies $(\gamma, \varepsilon)$-local DP.

The following conversion lemma from [13] relates Rényi DP to $(\varepsilon_{\mathsf{DP}}(\delta), \delta)$-DP.

**Lemma G.2.** If $\mathcal{A}$ satisfies $(\gamma, \varepsilon)$-Rényi DP for some $\gamma \geq 1$, then, for any $\delta > 0$, $\mathcal{A}$ satisfies $(\varepsilon_{\mathsf{DP}}(\delta), \delta)$-DP, where

$$\varepsilon_{\mathsf{DP}}(\delta) = \varepsilon + \frac{\log\left(1/\gamma\delta\right)}{\gamma-1} + \log(1 - 1/\gamma). \tag{23}$$

The following theorem states that, when simulating the Gaussian mechanism, PPR satisfies the following both central and local DP guarantee:

**Corollary G.3** (PPR-compressed Gaussian mechanism). *Let $\varepsilon \geq 0$ and $\gamma \geq 1$. Consider the Gaussian mechanism $P_{Z|X}(\cdot|x) = \mathcal{N}(x, \frac{\sigma^2}{n}\mathbb{I}_d)$, and the proposal distribution $Q = \mathcal{N}(0, (\frac{C^2}{d} + \frac{\sigma^2}{n})\mathbb{I}_d)$, where $\sigma \geq \sqrt{\frac{C\gamma}{2\varepsilon}}$. For each client $i$, let $Z_i$ be the output of PPR applied on $P_{Z|X}(\cdot|X_i)$. We have:*

- *$\hat{\mu}(Z^n) := \frac{1}{n}\sum_i Z_i$ yields an unbiased estimator of $\mu(X^n) = \frac{1}{n}\sum_{i=1}^n X_i$ satisfying $(\gamma, \varepsilon)$-(central) Rényi DP and $(\varepsilon_{\mathsf{DP}}(\delta), \delta)$-DP, where $\varepsilon_{\mathsf{DP}}(\delta)$ is defined in (23).*

- *$P_{Z|X_i}$ satisfies $(2\alpha\tilde{\varepsilon}_{\mathsf{DP}}(\delta), 2\delta)$-local DP, where*

$$\tilde{\varepsilon}_{\mathsf{DP}}(\delta) := \sqrt{n}\varepsilon + \frac{\log\left(1/\gamma\delta\right)}{\gamma-1} + \log(1 - 1/\gamma).$$

- *$\hat{\mu}(Z^n)$ has MSE $\mathbb{E}[\|\mu - \hat{\mu}\|_2^2] = \sigma^2 d/n^2$.*

- *The average per-client communication cost is at most $\ell + \log_2(\ell+1) + 2$ bits where*

$$\ell := \frac{d}{2}\log_2\left(\frac{C^2 n}{d\sigma^2} + 1\right) + \eta_\alpha \ \leq \ \frac{d}{2}\log_2\left(\frac{n\varepsilon^2}{2d\ln(1.25/\delta)} + 1\right) + \eta_\alpha,$$

*where $\eta_\alpha := (\log_2(3.56))/\min\{(\alpha-1)/2, 1\}$.*

**Proof.** The central DP guarantee follows from [68] and Lemma G.2. The local DP guarantee follows from Lemma G.2 and Theorem 4.8. Finally, the communication bound can be obtained from the same analysis as in Corollary 5.2. $\square$

# H Proof of Corollary 5.2

Consider the PPR applied on the Gaussian mechanism $P_{Z|X}(\cdot|x) = \mathcal{N}(x, \frac{\sigma^2}{n}\mathbb{I}_d)$, with the proposal distribution $Q = \mathcal{N}(0, (\frac{C^2}{d} + \frac{\sigma^2}{n})\mathbb{I}_d)$. PPR ensures that $Z_i$ follows the distribution $\mathcal{N}(X_i, \frac{\sigma^2}{n}\mathbb{I}_d)$. Therefore the MSE is

$$
\begin{aligned}
\mathbb{E}\left[\|\mu - \hat{\mu}\|_2^2\right] &= \mathbb{E}\left[\left\|\frac{1}{n}\sum_{i=1}^n (X_i - Z_i)\right\|_2^2\right] \\
&= \frac{1}{n} \cdot d \cdot \frac{\sigma^2}{n} \\
&= \frac{\sigma^2 d}{n^2}.
\end{aligned}
$$

For the compression size, for $x \in \mathbb{R}^d$ with $\|x\|_2 \le C$, we have

$$
\begin{aligned}
&D(P_{Z|X}(\cdot|x)\|Q) \\
&= \mathbb{E}_{Z \sim P_{Z|X}(\cdot|x)}\left[\log \frac{\mathrm{d}P_{Z|X}(\cdot|x)}{\mathrm{d}Q}(Z)\right] \\
&= \mathbb{E}_{Z \sim P_{Z|X}(\cdot|x)}\left[\log \frac{(2\pi\sigma^2/n)^{-d/2}\exp(-\frac{1}{2}\|Z-x\|_2^2/(\sigma^2/n))}{(2\pi(\frac{C^2}{d} + \frac{\sigma^2}{n}))^{-d/2}\exp(-\frac{1}{2}\|Z\|_2^2/(\frac{C^2}{d} + \frac{\sigma^2}{n}))}\right] \\
&= \mathbb{E}_{Z \sim P_{Z|X}(\cdot|x)}\left[\frac{d}{2}\log \frac{\frac{C^2}{d} + \frac{\sigma^2}{n}}{\sigma^2/n} + \frac{1}{2}\left(\frac{\|Z\|_2^2}{\frac{C^2}{d} + \frac{\sigma^2}{n}} - \frac{\|Z-x\|_2^2}{\sigma^2/n}\right)\right] \\
&\le \frac{d}{2}\log \frac{\frac{C^2}{d} + \frac{\sigma^2}{n}}{\sigma^2/n} + \frac{1}{2}\left(\frac{C^2 + \sigma^2 d/n}{\frac{C^2}{d} + \frac{\sigma^2}{n}} - \frac{\sigma^2 d/n}{\sigma^2/n}\right) \\
&= \frac{d}{2}\log\left(\frac{C^2 n}{d\sigma^2} + 1\right).
\end{aligned}
$$

Hence, by Theorem 4.3, the compression size is at most $\ell + \log_2(\ell + 1) + 2$ bits, where

$$
\begin{aligned}
\ell &:= \frac{d}{2}\log_2\left(\frac{C^2 n}{d\sigma^2} + 1\right) + \eta_\alpha \\
&\le \frac{d}{2}\log_2\left(\frac{n\epsilon^2}{2d\ln(1.25/\delta)} + 1\right) + \eta_\alpha \\
&\le \frac{n\epsilon^2 \log_2(e)}{4\ln(1.25/\delta)} + \eta_\alpha,
\end{aligned}
$$

where $\eta_\alpha := (\log_2(3.56))/\min\{(\alpha - 1)/2, 1\}$.

The central-DP guarantee follows from $(\varepsilon, \delta)$-DP of Gaussian mechanism [26, Appendix A] since the output distribution of PPR is exactly the same as the Gaussian mechanism, whereas the local-DP guarantee follows from Theorem 4.6 and [26, Appendix A].

# I Proof of Corollary 6.1

Let $\|X - Z\|_2 = RS$ where $R \in [0, \infty)$ is the magnitude of $X - Z$, and $\|S\|_2 = 1$. As shown in [33], $R$ follows the Gamma distribution with shape $d$ and scale $1/\varepsilon$. Hence the MSE is

$$
\mathbb{E}\left[\|X - Z\|_2^2\right] = \mathbb{E}\left[R^2\right] = \left(\frac{d}{\varepsilon}\right)^2 + \frac{d}{\varepsilon^2} = \frac{d(d+1)}{\varepsilon^2}.
$$

The conditional differential entropy (in nats) of $Z$ given $X$ is

$$h(Z|X) = h(R) + h(S|R)$$

$$= d + \ln \Gamma(d) - (d-1)\psi(d) - \ln \varepsilon + \mathbb{E}\left[\ln(nR^{d-1}\mathrm{Vol}(\mathcal{B}_d(1)))\right]$$

$$= d + \ln \Gamma(d) - (d-1)\psi(d) - \ln \varepsilon + \ln d + \ln(\mathrm{Vol}(\mathcal{B}_d(1))) + (d-1)\mathbb{E}\left[\ln R\right]$$

$$= d + \ln \Gamma(d) - (d-1)\psi(d) - \ln \varepsilon + \ln d + \frac{d}{2}\ln \pi - \ln \Gamma\left(\frac{d}{2} + 1\right)$$

$$\qquad - (d-1)\ln \epsilon + (d-1)\psi(d)$$

$$= d \ln \frac{e\sqrt{\pi}}{\varepsilon} + \ln \frac{d\Gamma(d)}{\Gamma(\frac{d}{2}+1)},$$

where $\psi$ is the digamma function. Therefore, the KL divergence between $P_{Z|X}(\cdot|x)$ and $Q$ (in nats) is

$$D(P_{Z|X}(\cdot|x)\|Q)$$

$$= -\mathbb{E}_{Z\sim P_{Z|X}(\cdot|x)}\left[\ln\left(\left(2\pi\left(\frac{C^2}{d} + \frac{d+1}{\varepsilon^2}\right)\right)^{-d/2}\exp\left(-\frac{\|Z\|_2^2}{2(\frac{C^2}{d}+\frac{d+1}{\varepsilon^2})}\right)\right)\right] - h(Z|X)$$

$$= \frac{d}{2}\ln\left(2\pi\left(\frac{C^2}{d} + \frac{d+1}{\varepsilon^2}\right)\right) + \frac{\mathbb{E}_{Z\sim P_{Z|X}(\cdot|x)}\left[\|Z\|_2^2\right]}{2(\frac{C^2}{d}+\frac{d+1}{\varepsilon^2})} - d\ln\frac{e\sqrt{\pi}}{\varepsilon} - \ln\frac{d\Gamma(d)}{\Gamma(\frac{d}{2}+1)}$$

$$\leq \frac{d}{2}\ln\left(2\pi\left(\frac{C^2}{d} + \frac{d+1}{\varepsilon^2}\right)\right) + \frac{C^2 + \frac{d(d+1)}{\varepsilon^2}}{2(\frac{C^2}{d}+\frac{d+1}{\varepsilon^2})} - d\ln\frac{e\sqrt{\pi}}{\varepsilon} - \ln\frac{d\Gamma(d)}{\Gamma(\frac{d}{2}+1)}$$

$$= \frac{d}{2}\ln\left(\frac{2}{e}\left(\frac{C^2\varepsilon^2}{d} + d + 1\right)\right) - \ln\frac{\Gamma(d+1)}{\Gamma(\frac{d}{2}+1)}.$$

Hence, by Theorem 4.3, the compression size is at most $\ell + \log_2(\ell + 1) + 2$ bits. The metric privacy guarantee follows from Theorem 4.7.

## J   Experiments on Metric Privacy

We use PPR to simulate the Laplace mechanism [3, 33, 34] $f_{Z|X}(z|x) \propto e^{-\varepsilon d_{\mathcal{X}}(x,z)}$ discussed in Section 6. We consider $X \in \mathcal{B}_d(C)$ where $C = 10000$ and $d = 500$. A large number of dimensions $d$ is common, for example, in privatizing word embedding vectors [33, 34]. We compare the performance of PPR-compressed Laplace mechanism (Corollary 6.1) with the discrete Laplace mechanism [3]. The discrete Laplace mechanism is described as follows (slightly modified from [3] to work for the $d$-ball $\mathcal{B}_d(C)$): 1) generate a Laplace noise $Y$ with probability density function $f_Y(y) \propto e^{-\varepsilon\|y\|_2}$; 2) compute $\hat{Z} = X + Y$; 3) truncate $\hat{Z}$ to the closest point $Z$ in $\mathcal{B}_d(C)$; and 4) quantize each coordinate of $Z$ by a quantizer with step size $u > 0$. The number of bits required by the discrete Laplace mechanism is $\lceil \log_2(\mathrm{Vol}(\mathcal{B}_d(C))/u^d)\rceil$, where $\mathrm{Vol}(\mathcal{B}_d(C))/u^d$ is the number of quantization cells (hypercube of side length $u$) inside $\mathcal{B}_d(C)$. The parameter $u$ is selected to fit the number of bits allowed.

Figure 2 shows the mean squared error of PPR-compressed Laplace mechanism ($\alpha = 2$) and the discrete Laplace mechanism for different $\varepsilon$'s, when the number of bits is limited to 500, 1000 and 1500.[16] We can see that PPR performs better for larger $\epsilon$ or smaller MSE, whereas the discrete Laplace mechanism performs better for smaller $\epsilon$ or larger MSE. The performance of discrete Laplace mechanism for smaller $\epsilon$ is due to the truncation step which limits $Z$ to $\mathcal{B}_d(C)$, which reduces the error at the expense of introducing distortion to the distribution of $Z$, and making $Z$ a biased estimate of $X$. In comparison, PPR preserves the Laplace conditional distribution $f_{Z|X}$ exactly, and hence produces an unbiased $Z$.

---

[16]The MSE of PPR is computed using the closed-form formula in Corollary 6.1, which is tractable since $Z$ follows the Laplace conditional distribution $f_{Z|X}$ exactly. The number of bits used by PPR is given by the bound in Corollary 6.1. The MSE of the discrete Laplace mechanism is estimated using 5000 trials per data point. Although we do not plot the error bars, the largest coefficient of variation of the sample mean (i.e., standard error of the mean divided by the sample mean) is only 0.00117, which would be unnoticeable even if the error bars were plotted.

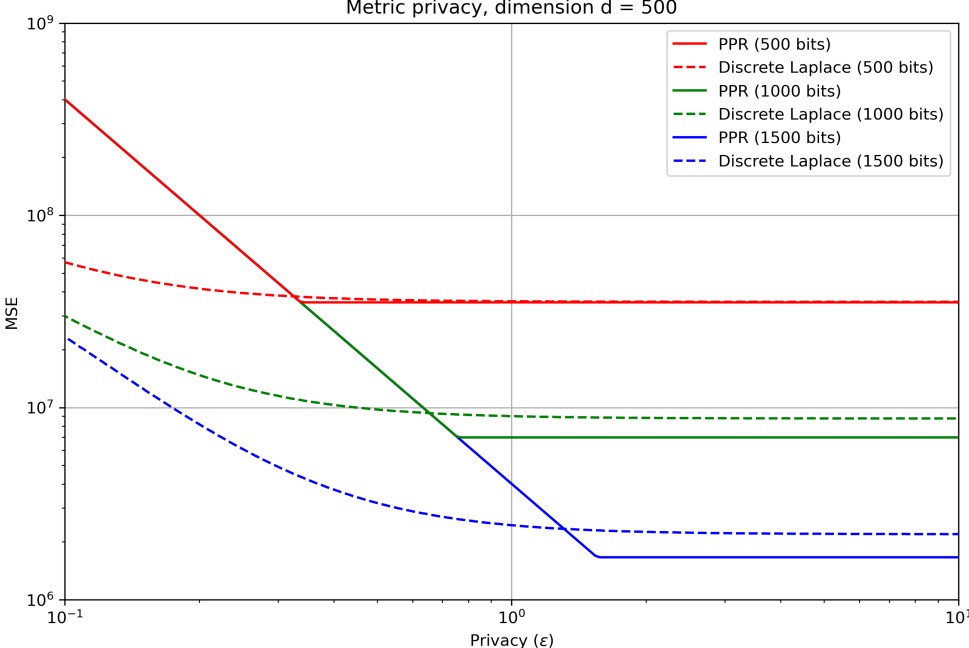

Figure 2: MSE of PPR-compressed Laplace mechanism and discrete Laplace mechanism [3] for different $\varepsilon$'s.

## K   Running Time of PPR

### K.1

As discussed in Section 7, we can ensure an $O(d)$ running time for the Gaussian mechanism by using the sliced PPR, where the $d$-dimensional vector $X$ is divided into $\lceil d/d_{\text{chunk}} \rceil$ chunks, each with a fixed dimension $d_{\text{chunk}}$ (possibly except the last chunk if $d_{\text{chunk}}$ is not a factor of $d$). The average total running time is $\lceil d/d_{\text{chunk}} \rceil T_{\text{chunk}}$, where $T_{\text{chunk}}$ is the average running time of PPR applied on one chunk.[17] Therefore, to study the running time of the sliced PPR, we study how $T_{\text{chunk}}$ depend on $d_{\text{chunk}}$.

In Figure 3 we show the running time $T_{\text{chunk}}$ of PPR applied on one chunk with dimension $d_{\text{chunk}}$, where $d_{\text{chunk}}$ ranges from 40 to 110.[18] With $d = 1000$, $n = 500$, $\varepsilon = 0.05$ and $\delta = 10^{-6}$, we require a Gaussian mechanism with noise $\mathcal{N}(0, n\tilde{\sigma}^2 \mathbb{I}_{d_{\text{chunk}}})$ where $\tilde{\sigma} = 1.0917$ at each user in order to ensure $(\varepsilon, \delta)$-central DP. We record the mean $T_{\text{chunk}}$ and the standard error of the mean[19] of the running time of PPR applied to simulate this Gaussian mechanism (averaged over 20000 trials).

---

[17]Note that the chunks may be processed in parallel for improved efficiency.

[18]Experiments were executed on M1 Pro Macbook, 8-core CPU ($\approx 3.2$ GHz) with 16GB memory.

[19]The standard error of the mean is given by $\sigma_{\text{mean}} = \sigma_{\text{time}}/\sqrt{n_{\text{trials}}}$, where $\sigma_{\text{time}}$ is the standard deviation of the running time among the $n_{\text{trials}} = 20000$ trials.

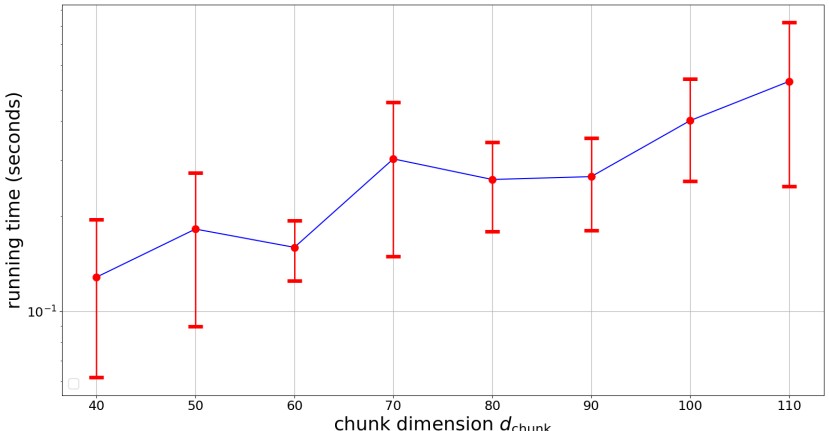

Figure 3: Average running time of PPR applied to a chunk of dimension $d_{\mathrm{chunk}}$, with error bars indicating the interval $T_{\mathrm{chunk}} \pm 2\sigma_{\mathrm{mean}}$, where $T_{\mathrm{chunk}}$ is the sample mean of the running time, and $\sigma_{\mathrm{mean}}$ is the standard error of the mean (see Footnote 19).

### K.2

We plot the average running time (over 20000 trials for each data point) against the values of $\epsilon \in [0.06, 10]$, with $d_{\mathrm{chunk}}$ always chosen to be 4. The average running time is denoted as $T_{\mathrm{chunk}}$, and the standard error of the mean is given by $\sigma_{\mathrm{mean}} = \sigma_{\mathrm{time}}/\sqrt{n_{\mathrm{trials}}}$, where $\sigma_{\mathrm{time}}$ is the standard deviation of the running time among the $\sigma_{\mathrm{time}} = 20000$ trials.

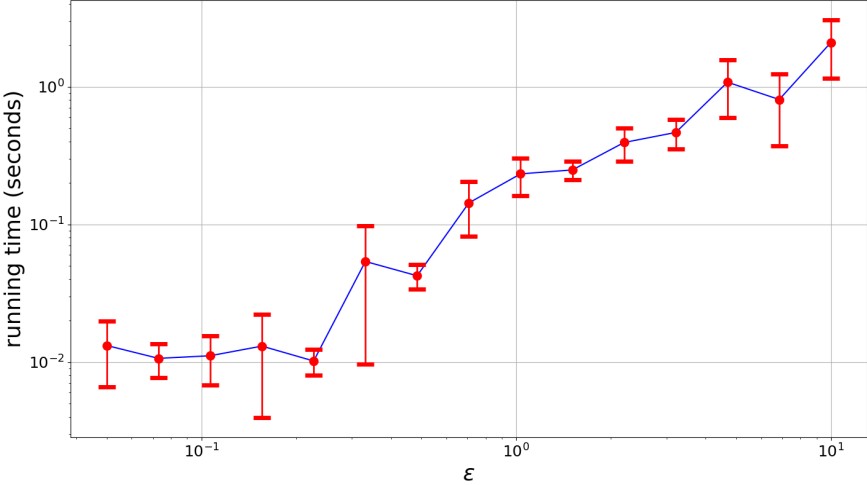

Figure 4: Average running time (over 20000 trials), $d_{\mathrm{chunk}} = 4$ and $\varepsilon \in [0.06, 10]$, with error bars indicating the interval $T_{\mathrm{chunk}} \pm 2\sigma_{\mathrm{mean}}$, where $T_{\mathrm{chunk}}$ is the sample mean of the running time, and $\sigma_{\mathrm{mean}}$ is the standard error of the mean.

## L    MSE against Compression Size

We plot the MSE against the compression size (ranging from 25 to 1000 bits) for $\epsilon \in \{0.25, 0.5, 1.0, 2.0\}$ in the following figure.

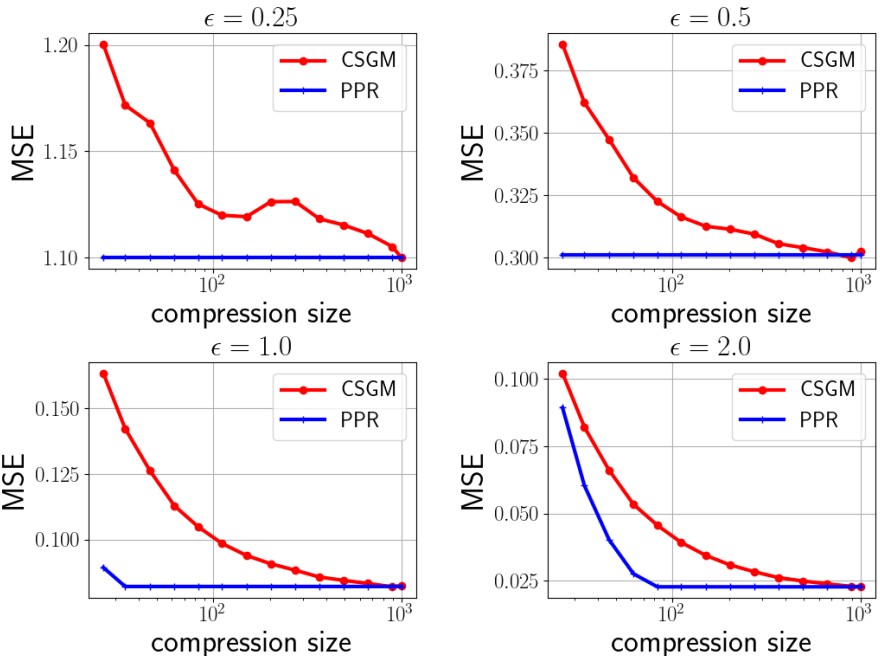

Figure 5: The MSE of PPR and CSGM against the compression size in bits, where $\varepsilon$ is chosen from $\{0.25, 0.5, 1.0, 2.0\}$ and compression sizes vary from 25 to 1000 bits. Note that parts of the curves for PPR are flat, because a lower compression size is already sufficient for PPR to exactly simulate the best Gaussian mechanism for that value of $\varepsilon$, so a higher compression size than necessary will not affect the result.

