# OpenReview forum: "Universal Exact Compression of Differentially Private Mechanisms"
_NeurIPS.cc/2024/Conference — NeurIPS 2024 poster_

### Official Review · Reviewer_GP8R · 2024-07-05

**Soundness:** 3
**Presentation:** 2
**Contribution:** 3
**Rating:** 7
**Confidence:** 2

**Summary:**

A new technique for compressing local differential privacy (LDP) reports is presented, based on Poisson functional representations, a tool from information theory that allows encoding a random variable in close to information-theoretically minimum number of bits (expected) in a "universal" manner that does not require the decoder to know about the distribution of the random variable. The paper presents a generalization called Poisson *private* representations (PPR) that preserves DP guarantees up to a multiplicative factor and has similar encoding efficiency (with a trade-off controlled by a parameter 𝛼). Unlike previous LDP compression techniques, PPR result in exactly the same distribution of decoded values as the original, uncompressed LDP protocol, so it preserves properties such as unbiasedness. The utility of PPR is investigated in a couple of case studies: distributed mean estimation and metric privacy.

**Strengths:**

* Originality: Interesting techniques from information theory that have not, to my knowledge, been used in differential privacy or federated learning before
* Quality and clarity: The paper is very well-written (but also dense and assumes a lot from the reader).
* Significance: A little hard to gauge, but the generality of the methods suggest that they may be of interest in many settings. The "universality" is particularly striking and may have important applications.

**Weaknesses:**

* I found the paper hard to read, in part because it uses math that I was not familiar with, in part because many details are hidden away in the appendices. Possibly, a more self-contained and accessible version could be achieved by focusing the exposition on a special case (e.g. simulating privUnit), and leaving the generalization for the appendices.
* It is assumed that each client and the server share a common, random sequence. While this might be realized using a pseudorandom generator, communicating a short seed, this overhead may be significant in practice. (This weakness, it should be mentioned, is shared with several previous methods.)
* PPR yields an overhead of more than a factor 2 in the privacy parameter
* The running time is exponential in 𝜺 which may limit applications to the shuffle model, where amplification means that high values of epsilon may be used

**Questions:**

* Is it correctly understood that your method is private even to an adversary who knows the shared randomness?
* Is there any way that clients can use the same shared randomness, or should this by per-client?
* In line 166 it is not clear that argmin is well-defined, can you comment on this?
* Can you achieve a bound on the worst-case encoding size, say, by reverting to a simpler encoding if K is too large?

**Limitations:**

The authors have adequately addressed limitations

---

> ### Author Rebuttal · Authors · 2024-08-07
>
> We thank Reviewer GP8R for the constructive feedback.
> We are pleased to hear Reviewer GP8R thought our PPR technique is interesting and the paper is very well-written.
> Please find our responses to the questions and comments below.
>
> **Regarding a more self-contained and accessible version on a special case:** In the revised manuscript, we will explain the PPR algorithm using the Gaussian mechanism as an example, which will hopefully make the exposition more accessible.
>
>
> **Regarding the common random sequence:** In practice, the common random sequence is generated using a pseudorandom number generator (PRNG) initialized with a seed shared between the client and the server. While this seed must be communicated between the client and the server, and may lead to an increased communication cost, note that the client and the server only ever need to communicate one seed, which will be used to initialize a PRNG that will be used in all subsequent privacy mechanisms and communication tasks. This is similar to running a large-scale randomized simulation program on a computer, where we only need to initialize the random seed once when the simulation starts.
>
> Practically, the client and the server will share the seed when the connection is established as a small overhead, and can use the same PRNG throughout the whole connection. If the client is applying DP mechanisms to transmit a high-dimensional data, or is using DP mechanisms many times, the cost of communicating the seed will only contribute a small fraction of the total communication cost. On the other hand, if the client is only applying DP mechanisms a small number of times on some small data, then the cost of communicating the seed will be dominated by the overhead in TCP/IP handshaking. In any case, the cost of communicating the seed is insignificant.
>
>
>  **Regarding running time for larger $\\varepsilon$:** We have discussed the running time of case $\\varepsilon = 0.05$ in the paper.
> We report the running time for some larger values of $\\varepsilon$'s as follows:
> For $\\varepsilon = 6$ (which is the largest $\\varepsilon$ that is plotted in Figure 1), we can choose $d_{\\mathrm{chunk}}=2$ in the sliced PPR to have an average running time $0.0127$ seconds or choose $d_{\\mathrm{chunk}}=4$ to have an average running time $0.6343$ seconds, where we calculate the running time by averaging over $10000$ trails.
> For $\\varepsilon = 10$ (as suggested by the reviewer),  we can choose $d_{\\mathrm{chunk}}=2$ to have an average running time $0.0128$ seconds or choose $d_{\\mathrm{chunk}}=4$ to have an average running time $0.7301$ seconds.
> We plot the average running time (over $10000$ trials for each data point) against $\\varepsilon\\in [0.06, 10]$, with $d_{\\mathrm{chunk}}$ fixed to $4$.
> Please refer to Figure A in the attached pdf file, where we record the mean $T_{chunk}$ and the standard error of the mean.
> The standard error of the mean is given by $\\sigma_{\\mathrm{mean}} = \\sigma_{\\mathrm{time}} / \\sqrt{n_{\\mathrm{trials}}}$, where $\\sigma_{\\mathrm{time}}$ is the standard deviation of the running time among the $n_{\\mathrm{trials}}=10000$ trials.
>
>
> **Regarding the overhead in the privacy parameter:** For $\\varepsilon$-DP, PPR may inflate the privacy budget $\\varepsilon$ by a factor of $2 \\alpha$ as shown in Theorem 4.5 (though we can make it arbitrarily close to $2$). However, if we instead consider $(\\varepsilon, \\delta)$-DP with a small $\\delta$, then Theorem 4.8 shows that the privacy budget $\\varepsilon$ of the compressed mechanism can be arbitrarily close to the $\\varepsilon$ of the original mechanism.
>
>
> **Regarding exponential running time:** We agree with the reviewer that we can discuss more about the running time in main sections.
> We will move the paragraph "the running time complexity (which depends on the number of samples $Z_i$ the algorithm must examine before outputting the index $K$) can be quite high. Since $\\mathbb{E}[\\log K] \\approx I(X;Z)$, $K$ (and hence the running time) is at least exponential in the mutual information $I(X;Z)$" from the limitation section to after Theorem 4.3.
>
>
> **Regarding shared randomness and privacy:** All privacy analyses in the paper assumes that the adversary knows both the message $K$ and the shared randomness $(Z_i)_i$. The trade-off between communication cost and privacy are described in Theorem 4.3 and Theorems 4.5-4.8.
>
>
> **Regarding clients using the same shared randomness:** It is assumed that each client uses a different independent shared randomness. Otherwise, we can no longer ensure that the privacy-preserving noises at the clients are independent. Nevertheless, as mentioned earlier in this response, the cost of generating these shared randomness is insignificant. We will clarify this in the revised manuscript.
>
>
> **Regarding the argmin in line 166:** Since the $T_i$'s are continuous random variables, with probability one, there do not exist two equal values among $\\tilde{T}_i$'s. This will be clarified in the revised manuscript.
>
>
> **Regarding worst-case encoding size:** We can apply Markov inequality $\\mathrm{Pr}(\\log K > L) \\le \\mathrm{E}[\\log K] / L$ on the bound in Theorem 4.3 to show that $\\log K$ is most likely small. If the worst-case encoding length must be controlled, i.e., $\\log K \\le L$, we can modify the method to have the encoder output $0$ instead if the $K$ given by PPR exceeds $2^L$. After this modification, the method may not be exact, though the error probability is bounded by $\\mathrm{Pr}(\\log K > L)$, which is small due to Markov inequality and the bound in Theorem 4.3. We will explain this in the revised paper.

---

> > ### Comment · Reviewer_GP8R · 2024-08-08
> > **Thanks for the rebuttal**
> >
> > I have no further questions

---

> > > ### Author Response · Authors · 2024-08-14
> > > **Thank you for your response to our rebuttal.**
> > >
> > > We thank the reviewer for the time reading our paper and rebuttal.

---

### Official Review · Reviewer_U6Tv · 2024-07-08

**Soundness:** 3
**Presentation:** 2
**Contribution:** 2
**Rating:** 7
**Confidence:** 3

**Summary:**

The paper addresses the problem of reducing the communication cost of messages that are shared under differential privacy (DP) guarantees. This is an important problem in privacy preserving machine learning, where parties that share obfuscated large models could incur in significant communication overhead.

Techniques to reduce communication exist, but they (a) require to share randomness between the encoder (i.e., the data subject that obfuscates its data) and a (possibly adversarial) analyst (or decoder), (b) resort into approximate and biased distribution estimations or (c) only work for specific distributions.

The current paper proposes a compression technique that (i) achieves optimal and lossless compression rates without significant degradation of DP-guarantees, (ii) only requires a small amount of public information to be shared between encoder and decoder and (iii) are universal as they can be used for any DP-mechanism.

**Strengths:**

In terms of novelty, the paper proposes an original idea that leverages the Poisson functional representation to avoid shared randomness between the encoder and decoder, allowing the use of the compression technique in the local DP setting.

The work is of good quality as
- it appropriately backs its claims with proofs
- provides an extensive analysis of the compression and privacy properties of the proposed technique.

First, it shows that it achieves optimal compression (up to a logarithmic factor) which yields to compression rates that are similar to non-private techniques. Next, it provides DP bounds of the compression technique, characterizing the trade-offs between compression and privacy and showing that significant compression can be achieved with acceptable privacy degradation. Finally, it shows how these guarantees boil down to distributed mean estimation (DME) and metric privacy.

A strong point on the impact of the contribution is the universality property, which allows the simulation of any distribution. This makes the technique widely applicable, unlike previous solutions that are tailored to specific problems.

**Weaknesses:**

The main drawback of the contribution is in the lack of a more detailed comparison with other previous techniques. While the proposed technique successfully positions itself with respect to related work, it is not clear how the advantages (universality, exact simulation, lack of shared randomness between encoder and decoder) lead to significant improvements in machine learning.

As said before, it is clear that universality is a potentially interesting property. However, it is not completely clear the disadvantage of shared randomness when these randomness does not break the privacy guarantee (as it seems to be the case in [5,30]). It seems that this shared randomness would imply a larger communication cost, but the final compromises between compression, accuracy and privacy are not clear.

The advantages of an exact simulation are also not illustrated. It might be the case that the degradation of accuracy (or privacy) of approximate solutions is not significant. Since these aspects are not more accurately illustrated, I am not sure about the actual advantages of the current technique.

Finally, empirical illustrations don't to show more than a marginal advantage with respect to GSGM [18], which only apply in permissive privacy regimes (e.g. $\epsilon > 1$). Therefore, these illustrations do not help to clarify the advantages of the current contribution in practice.

A minor comment is the lack of a more complete presentation of Poisson functional representation, which could be expanded for clarity.

**Questions:**

Can de authors address weaknesses raised in my review?

Especially, two aspects that would clarify the advantages of the work are:

- The disadvantages of shared randomness between encoder and decoder, which in the paper is discussed in two different contexts: At the end page 2 of the paper, shared randomness appear to break privacy guarantees if the decoder is an adversary. Then, at the beginning of page 3 under the compression of local DP mechanisms, shared randomness is also discussed, but disadvantages here are not clear (see the related point in the weaknesses I described in my review).

- From Figure 1, I can only see the advantages of PPR in permissive privacy regimes. However, distributed mean estimation is a canonical task that in machine learning is used under composition, leading to further privacy degradation. Therefore it is important to understand the properties of the protocol under more conservative regimes (e.g. $\epsilon < 1/2$). Could PPR perform better with respect to CSGM with further compression (e.g., 25 bits or 10 bits) under such regimes?

**Limitations:**

I don't see additional limitations other than the discussed by the authors.

---

> ### Author Rebuttal · Authors · 2024-08-07
>
> We thank Reviewer U6Tv for the constructive and detailed feedback. We are pleased to hear that Reviewer U6Tv appreciate the universality of our proposed method. Please find our responses to the comments below.
>
> **Shared randomness and privacy:** Whether shared randomness weakens or prevents privacy depends on the design of the mechanism. Indeed, the 1-bit protocol [5] guarantees privacy in the presence of shared randomness. [30] is slightly different since its privacy-utility trade-off depends on computational assumptions, which is not the case for our work (and most related works cited by us). In some other algorithms, shared randomness can be detrimental to privacy, and additional steps or assumptions are needed to alleviate this problem (e.g., [43] requires a trusted aggregator; [46] requires secure aggregation).
>
> Note that our proposed PPR algorithm also requires shared randomness (see Section 4; it will be made clearer in the revised version), though it is designed to ensure privacy even when the adversary can access both the shared randomness and the compressed data. All privacy analyses (Theorems 4.5-4.8) in the paper assume that the adversary knows both the message $K$ and the shared randomness $(Z_i)_i$.
>
> To clarify, using shared randomness is not an advantage or a disadvantage by itself. It only becomes a disadvantage if it harms the privacy, which may happen in dithered quantization schemes (unless with additional steps and/or assumptions [43,46]), but does not happen in [5,30,65] or our PPR algorithm. The purpose of mentioning shared randomness in the introduction and related work section is to highlight the challenges in ensuring privacy in the presence of shared randomness, though we understand that it could give an impression that we are claiming shared randomness to be an advantage of the proposed algorithm, which is not our intention. This will be clarified in the revised version. The advantages of our algorithm are universality, exactness (comparing to [5,30] which are approximate) and communication efficiency.
>
> **The presentation of Poisson functional representation:** We will have a more detailed explanation of the Poisson functional representation in the revised version.
>
>
> **Advantages of exact simulation:**
> The advantages of exact simulation (refer to the ``Exactness'' paragraph in page 2) are:
>
> 1. Exact simulation does not introduce any bias in compression, and hence it guarantees unbiasedness for tasks such as distributed mean estimation (DME).
>
> 2. For the Gaussian mechanism for DME, guaranteeing that the resultant local noise of the compression is exactly Gaussian will result in an overall noise that is Gaussian as well. This provides the central DP of the PPR-simulated Gaussian mechanism, in addition to (and with a better $\\varepsilon$ than) the local DP. Otherwise, only a much looser central DP can be provided, since when the local noise is not "summable", one must rely on generic privacy amplification techniques, e.g. shuffling [Erlingsson et al., 2019], which suffers from highly sub-optimal constants and are only meaningful for limited privacy regimes where $\\varepsilon_{local} \\ll 1$, making it less practical for most FL applications.
>
> This highlights an important advantage of exactness. If the goal is only to design a stand-alone privacy mechanism, then we can study the privacy and utility of the mechanism, without studying the output distribution. However, if the output of the mechanism is used for downstream tasks (e.g., for DME, after receiving information from clients, the server sends information about the aggregated mean to data analysts, where central DP is crucial), having an exact characterization of the conditional distribution of the output given the input will allow us to obtain precise (central) privacy and utility guarantees. Otherwise, we must fall back to the worst-case guarantee (for DME, the central DP guarantee can only be the same as the local DP, which is far from optimal), or use a sub-optimal generic privacy amplification technique.
>
> **Advantages of PPR under more conservative regimes or with further compression:** The y-axis of Figure 1 is in logarithmic scale, which may make the MSE's look closer than they actually are. For example, when $\\varepsilon=1$ and we compress $d=1000$ to $50$ bits, CSGM has an MSE $0.1231$, while PPR has an MSE $0.08173$, giving a 33.61% reduction. For a case with further compression under more conservative $\\varepsilon$, for example, when $\\varepsilon=0.5$ and we compress $d=1000$ to $25$ bits, CSGM has an MSE $0.3877$, while PPR has an MSE $0.3011$, giving a 22.33% reduction. Such reductions are significant, considering that all considered mechanisms are asymptotically close to optimal, so a large improvement compared to an (almost optimal) mechanism is unexpected. We plot the MSE against the compression size (ranging from 25 to 1000 bits) for $\\varepsilon\\in \\{0.25, 0.5, 1.0, 2.0\\}$ in Figure B in the submitted pdf file.
>
> Moreover, we note again that PPR achieves a better trade-off between MSE and central DP than CSGM, while also giving local DP guarantees that CSGM cannot provide. Another advantage of PPR under more conservative regimes (small $\\varepsilon$) is that the trade-off between $\\varepsilon$ and MSE of PPR exactly coincides with the trade-off of the Gaussian mechanism for small $\\varepsilon$, as seen in Figure 1. In contrast, CSGM is close to (but strictly worse than) the Gaussian mechanism. This means that for small $\\varepsilon$, PPR provides compression without any drawback in terms of $\\varepsilon$-MSE trade-off compared to the Gaussian mechanism (which requires an infinite size communication to exactly realize). This advantage is a consequence of exact simulation.
>
>
> **References**
>
> [Erlingsson, Úlfar, et al.] ``Amplification by shuffling: From local to central differential privacy via anonymity,'' SODA 2019.

---

> > ### Comment · Reviewer_U6Tv · 2024-08-13
> >
> > Thank you for your rebuttal and your efforts for clarifying concerns.
> >
> > I appreciate the clarifications on the use of shared randomness and the advantages with respect to CSGM [18]. You have addressed these points successfully.
> >
> > However, I still feel that the proposed technique should be further compared with related work. It is not clear to me why [18] is the only work for which you should compare concrete trade-offs. Even if you have been clear about the conceptual advantages of exact simulation (i.e., unbiased distributions, "summable" noise), it does not seem clear why approximate simulation would imply biased estimates. Therefore, I don't fully understand why other techniques mentioned in "Generic compression of local DP mechanisms" and "DME under DP" paragraphs of Section 2 are excluded from more concrete comparisons made in Section 7.
> >
> > In particular (and as already raised in the discussion), the proposed protocol inflates the privacy budget. Therefore if we want to obtain a compressed mechanisms with the exact same $(\epsilon, \delta)$ parameters as the original, your protocol should simulate a distribution that provides privacy with lower privacy budget, modifying (even if probably slightly) the distribution parameters. I don't see why this is not a distortion of the original distribution as claimed for other techniques. Therefore, my impression is that the work should either compare with a more extensive number of techniques, putting more illustrations of concrete privacy-accuracy-communication trade-offs on the table or better clarify why these comparisons are not relevant.

---

> > > ### Author Response · Authors · 2024-08-14
> > >
> > > We thank Reviewer U6Tv for the thoughtful comments. Please find our responses below.
> > >
> > > **1. The reason for comparing to CSGM [18]**
> > >
> > > We chose to compare to CSGM [18] since [18] is asymptotically optimal and compares favorably to several previous algorithms, as demonstrated in the experiments in [18]. By showing that our proposed scheme compares favorably to [18], we show that our scheme compares favorably to those previous works as well.
> > >
> > > **2. Regarding comparison to other techniques**
> > >
> > > Regarding Feldman and Talwar [30]: [30] mentioned applying [30, Theorem 3.4] to Gaussian mechanism. The main obstacle in comparing [30, Theorem 3.4] to our result is that [30] relies on a computational hardness assumption on the pseudorandom number generator, and it is unclear how many bits of random seed (communication) are necessary to guarantee computational indistinguishability. Also, [30] does not prove that their scheme has a communication cost close to the theoretical minimum.
> > >
> > > Regarding Bassily and Smith [5]: [5] does not exactly preserves the distribution of the simulated mechanism (there is a 50% chance that the data is dropped). Additional (likely non-trivial) analyses are necessary to characterize its central-DP guarantee for mean estimation, to be compared to our scheme in Figure 1.
> > >
> > > [65] is also non-exact, making their central-DP guarantees unclear.
> > >
> > > **3. About whether "approximate simulation would imply biased estimates"**
> > >
> > > We emphasize that unbiasedness is a mathematical property that requires proof. If an approximate method cannot be proved to be unbiased, then it should not be considered to be unbiased, regardless of how close to zero the bias appears to be in experiments (the bias might become large outside of the cases experimented). For some specific approximate methods, it may be possible to add a debiasing step with a proof of unbiasedness (e.g., [65]), though the feasibility of such a step depends on the specific task and the mathematical tractability of the output distribution. The advantage of exactness is that it can readily imply unbiasedness, and no additional steps are necessary.
> > >
> > > In sum, approximate method does not necessarily imply biasedness (e.g., [65]), but an approximate method without a proof of unbiasedness should not be considered to be unbiased. Therefore, unbiasedness is an advantage of our exact method over other approximate methods without proofs of unbiasedness. This is a theoretical advantage that does not require experiments to show.
> > >
> > > **4. Regarding distortion for a fixed $(\\varepsilon, \\delta)$ and approximate techniques**
> > >
> > > Indeed, if $(\\varepsilon, \\delta)$ is fixed, then we have to reduce the $\\varepsilon, \\delta$ of the simulated mechanism. However, this reduction can be small according to Theorem 4.8. Also, the distortion introduced by lowering $\\varepsilon, \\delta$ is considerably different from the distortion introduced by the methods in [30,65,71]. If we exactly simulate the Gaussian mechanism with a lower $\\varepsilon, \\delta$, the noise is still exactly Gaussian (with a larger variance), and can be added to other Gaussian noises nicely. This "summable" noise is the key to providing central DP guarantees (in addition to local DP). However, if we simulate the Gaussian mechanism using the approximate techniques in [30,65,71], the noise introduced has a mathematically intractable distribution, which will be an obstacle in obtaining theoretic guarantees for downstream tasks (e.g., for distributed mean estimation, we do not know the overall noise distribution and its central-DP properties after summing all the data). The main benefits of exactness are mathematical tractability and ease of proving guarantees for downstream tasks.
> > >
> > > **5. The benefits of simple theoretical guarantees**
> > >
> > > Also, we believe that the theoretical contributions of our work, namely the privacy-communication trade-off given in Theorems 4.3-4.8 given in simple and clean expressions, with exact preservation of the output distribution, is noteworthy in itself. To the best of our knowledge, our work is the first method for compressing DP mechanisms that has a bound on its compression size universally close to the I(X;Z) lower bound (i.e., the bound depends on the simulated mechanism only through I(X;Z) as in Theorem 4.3, and hence is always almost-optimal regardless of the situation). Even though we agree that more experiments can be beneficial, experiments are not strictly necessary to demonstrate the almost-optimality of our method, when we have a mathematical proof of its universal almost-optimality for the compression size.
> > >
> > > We believe these simple guarantees (in terms of simple quantities like I(X;Z)) and exact distribution preservation can make the proposed method a useful general technique for designing more specific DP mechanisms in the future.
> > >
> > >
> > > We hope that we have adequately addressed the concerns and questions, and kindly invite the reviewer to consider updating the score.

---

> > > > ### Comment · Reviewer_U6Tv · 2024-08-14
> > > >
> > > > Dear authors,
> > > >
> > > > Thank you for your reply. I highly appreciate your efforts in clarifying concerns. I am leaning to increase my score, but in that case I will do it after discussing with the other reviewers.

---

> > > > > ### Author Response · Authors · 2024-08-14
> > > > > **Thank you for your response to our rebuttal.**
> > > > >
> > > > > We thank the reviewer for the time reading our rebuttal and providing constructive feedbacks.

---

### Official Review · Reviewer_VD1w · 2024-07-12

**Soundness:** 3
**Presentation:** 3
**Contribution:** 3
**Rating:** 6
**Confidence:** 3

**Summary:**

The paper investigates the problem of compressing the output of differentially private algorithms, particularly focusing on the local model. Given a Local Differential Privacy (LDP) algorithm $ A $ that induces a conditional distribution $ p_{Z \mid X} $ where $ Z = A(X) $, the objective is to generate a message using minimum bits, allowing the server to recreate a random variable following the distribution $ p_{Z \mid X} $.

The paper introduces a novel compression algorithm leveraging shared randomness between the server and clients. Key points include:
1. The compression algorithm applies universally to every local randomized algorithm, provided there is agreement on the output space.
2. The size of the compressed output matches the mutual information lower bound $ I(X; Z) $, with an additive logarithmic term.
3. Using the compressed outputs, the server can reconstruct random variables that exactly follow the distributions of the outputs of the local randomizers of the clients.

A central component of this approach is the Poison functional representation, a channel simulation scheme. Initially not differentially private, the paper extends it to ensure differential privacy using a technique akin to the exponential mechanism.

Experimental results demonstrate improved performance of the proposed algorithm compared to previous approaches.

**Strengths:**

1. Viewing the problem of compressing the output of Local Differential Privacy (LDP) algorithms as a channel simulation problem is inspiring, as it allows leveraging insights and results from the extensive study of channel simulation.

2. The proposed algorithm is versatile and applicable to a wide range of LDP algorithms, effectively reducing their communication costs to nearly optimal levels. It can serve as a fundamental primitive: LDP algorithm designers can focus on the privacy-utility trade-off without significant concerns about communication costs.

**Weaknesses:**

See ``Limitations''

**Questions:**

1. On page 2, line 75, "lower bound I(X;Y)" - - > "lower bound I(X;Z)".

2. In the experiment section, the paper only mentions the running time for $ \epsilon = 0.05 $ (line 335). It would be informative to include the running time for larger values of $ \epsilon $ experimented in the paper, such as $ \epsilon = 10 $. Specifically, it would be useful to know how does the running time scale with $ \epsilon $.

**Limitations:**

1.	1. The running time of the compression algorithm is exponential in the mutual information $ I(X; Z) $. While this aspect is crucial, it is currently mentioned in the limitation section rather than explicitly in the main theorem describing the properties of the compression algorithm.

2.	2. The proposed compression algorithm does not strictly preserve the privacy guarantee of the Local Differential Privacy (LDP) algorithm. Instead, it can inflate the privacy guarantee by a constant factor: an $ \epsilon $-LDP algorithm, after compression, might become $ 2\alpha \epsilon $-LDP, where $ \alpha > 1 $. This trade-off is acceptable when focusing on the asymptotic privacy-utility trade-off of LDP algorithms. However, for LDP algorithms that achieve optimal errors with optimal constants, applying the compression algorithm may result in algorithms with suboptimal constants.

---

> ### Author Rebuttal · Authors · 2024-08-07
>
> We thank Reviewer VD1w for the constructive feedback. We are pleased to hear that Reviewer VD1w find the idea of viewing the compression of the output of LDP algorithms as channel simulation inspiring, and our technique can serve as a fundamental primitive to DP algorithm designers. Please find our responses to the questions and comments below.
>
>
> **Regarding running time for larger $\\varepsilon$:** We have discussed the running time of case $\\varepsilon = 0.05$ in the paper.
> We report the running time for some larger values of $\\varepsilon$'s as follows:
> For $\\varepsilon = 6$ (which is the largest $\\varepsilon$ that is plotted in Figure 1 of the original manuscript), we can choose $d_{\\mathrm{chunk}}=2$ in the sliced PPR to have an average running time $0.0127$ seconds or choose $d_{\\mathrm{chunk}}=4$ to have an average running time $0.6343$ seconds, where we calculate the running time by averaging over $10000$ trails;
> for $\\varepsilon = 10$ (as suggested by the reviewer),  we can choose $d_{\\mathrm{chunk}}=2$ to have an average running time $0.0128$ seconds or choose $d_{\\mathrm{chunk}}=4$ to have an average running time $0.7301$ seconds.
> It shows the running time are acceptable even for large values of $\\varepsilon$.
> We plot the average running time (over $10000$ trials for each data point) against the values of $\\varepsilon\\in [0.06, 10]$, with $d_{\\mathrm{chunk}}$ always chosen to be $4$, in Figure A in the submitted pdf file.
>
> **Regarding exponential running time:** We agree with the reviewer that we can discuss more about the running time in main sections.
> We will move the paragraph ``the running time complexity (which depends on the number of samples $Z_i$ the algorithm must examine before outputting the index $K$) can be quite high. Since $\\mathbb{E}[\\log K] \\approx I(X;Z)$, $K$ (and hence the running time) is at least exponential in the mutual information $I(X;Z)$'' from the limitation section to after Theorem 4.3.
>
> **Regarding privacy guarantee:** For $\\varepsilon$-DP, PPR may inflate the privacy budget $\\varepsilon$ by a factor of $2 \\alpha$ as shown in Theorem 4.5 (though we can make it arbitrarily close to $2$). However, if we instead consider $(\\varepsilon, \\delta)$-DP with a small $\\delta$, then Theorem 4.8 shows that the privacy budget $\\varepsilon$ of the compressed mechanism can be arbitrarily close to the $\\varepsilon$ of the original mechanism.
>
> Moreover, we would like to note again that another advantage of our exact simulation scheme is that PPR enables having both local and central DP guarantees at the same time.

---

### Official Review · Reviewer_NbQx · 2024-07-12

**Soundness:** 3
**Presentation:** 3
**Contribution:** 2
**Rating:** 7
**Confidence:** 3

**Summary:**

The paper designs Poisson private representation (PPR) to compress and simulate any local randomizer while ensuring local differential privacy. PPR exactly preserves the joint distribution of the data and the output of the original local randomizer, and also achieves a compression size within a logarithmic gap from the theoretical lower bound. The authors also a provides an order-wise trade-off between communication, accuracy, central and local differential privacy for distributed mean estimation. Numerical experiments are conducted to validate theoretical justifications.

**Strengths:**

* The paper designs a compressor that can simulate any local or central DP mechanism and enables exact simulation
* The trade-off between communication, accuracy, and privacy for distributed mean estimation are analyzed
* The paper is well-organized and easy to follow
* The comparisons with previous results and the limitations of the proposed PPR are discussed

**Weaknesses:**

The privacy analysis in Corollary 5.2 is only suitable for cases where n, the number of local clients, is small. However, in practical scenarios such as federated learning, n is typically very large. An analysis that addresses privacy for large n would be appreciated.

**Questions:**

Why does the privacy analysis in Corollary 5.1 not relate to n?

**Limitations:**

Limitations are clearly stated in the paper.

---

> ### Author Rebuttal · Authors · 2024-08-07
>
> We thank Reviewer NbQx for the constructive feedback.
> We are pleased to hear Reviewer NbQx thought our manuscript is well-organized and easy to follow.
> Below, we clarify the weakness and address the question pointed out by the reviewer
>
>
> **Regarding small $n$:** Firstly, as noted in footnote 6, the restriction $\\varepsilon < 1/\\sqrt{n}$ (which may require a smaller $n$) is due to the simpler privacy accountant [24]. A tighter result that applies to any $n$ can be obtained by considering Rényi DP instead, as discussed in Corollary G.3 in Appendix G.
>
> Moreover, in the context of federated learning or analytics, $n$ refers to the cohort size, which is the number of clients *in each round*. This cohort size is typically much smaller than the total number of available clients. For example, as observed in [Kairouz et al., 2019], the per-round cohort size in Google's FL application typically ranges from $10^3$ to $10^5$, which is significantly smaller than the number of trainable parameters $d \\in [10^6, 10^9]$ or the number of available users $N \\in [10^6, 10^8]$.
>
> Furthermore, even in the context of traditional, non-FL training with a DP-SGD-type optimizer, $n$ refers to the *batch* size rather than the total number of samples, where privacy can be amplified via random batching.
>
> **Regarding the reason for which Corollary 5.1 does not relate to $n$:** Note that Corollary 5.1 (and also PrivUnit) considers local DP instead of central DP (see Definition 4.1). Therefore, the privacy analysis and guarantees do not depend on the total number of clients $n$.
>
>
> **References**
>
> [Kairouz et. al, 2019] "Advances and open problems in federated learning."

---

> > ### Comment · Reviewer_NbQx · 2024-08-13
> >
> > I appreciate the authors' thorough responses to my questions. I will keep my positive score.

---

> > > ### Author Response · Authors · 2024-08-14
> > > **Thank you for your response to our rebuttal.**
> > >
> > > We thank the reviewer for the time evaluating our manuscript.

---

### Author Rebuttal · Authors · 2024-08-07

Dear Reviewers and ACs,

We would like to thank all the reviewers for carefully reviewing our paper, their patience and also their valuable and constructive feedback.
We observed that the feedback from all four reviewers is generally positive.
Most reviewers mentioned the novelty of introducing a variant of the Poisson functional representation (as an elegant channel simulation technique) to compressing differentially private algorithms, our theoretic contributions with experimental validation and our organized structure among the strengths of our work.

The reviewers also shared suggestions that are valuable to improve our manuscript's quality.
We have responded to every question in our separate responses to each reviewer.
We also found that some concerns are common and mentioned by multiple reviewers, including the inflated privacy guarantee, the shared randomness, the running time and the advantage of exact simulation.
Hence we briefly summarize our replies to them here for a general clarification.
More details can be found in separate responses to each reviewer.

**Regarding shared randomness:**
All our privacy analyses assumes that the adversary knows both the message and the shared randomness.
In practice, shared randomness does not require a large communication cost.
The client and the server can communicate a small random seed to initialize the pseudorandom number generator (PRNG) that produces all the shared randomness needed.
Note that the client-server pair only ever need to communicate one seed for all subsequent privacy mechanisms and communication tasks.
In practice, the cost of communicating the seed is insignificant compared to both the cost of transmitting the compressed data (e.g., the $K$ in the proposed method), and the overhead of initializing the connection (e.g., TCP/IP handshaking).

**Regarding privacy guarantee:**
For $\\varepsilon$-DP, the PPR may inflate the privacy budget $\\varepsilon$ by a factor of $2 \\alpha$ (that can be arbitrarily close to $2$). However, we can instead consider $(\\varepsilon, \\delta)$-DP with a small $\\delta$, and Theorem 4.8 guarantees that the privacy budget $\\varepsilon$ of the compressed mechanism can be arbitrarily close to the $\\varepsilon$ of the original mechanism (i.e., almost no inflation).
We would like to note again that another advantage of PPR is the exact simulation, with advantages elaborated as follows.

**Regarding the exact simulation:**
Exact simulation (where the compression does not introduce any bias) enables us to guarantee 1) unbiasedness for tasks such as distributed mean estimation, where unbiasedness is crucial; 2) for the Gaussian mechanism for distributed mean estimation, the resultant local compression noise is exactly Gaussian, giving an overall noise that is also Gaussian so that we have both central DP and local DP.
Otherwise, without exact simulation we can only rely on generic privacy amplification techniques, such as shuffling [Erlingsson et al., 2019] (known to suffer from highly sub-optimal constants and only meaningful for limited privacy regimes where $\\varepsilon_{local} \\ll 1$), giving a much looser central DP.
Please see more discussions in our separate responses.

**Regarding exponential running time for larger $\\varepsilon$:**
We have discussed the running time of case $\\varepsilon = 0.05$ in the paper.
We report the running time (averaging over $10000$ trails) for some larger $\\varepsilon$'s:
For $\\varepsilon = 6$, by using sliced PPR with $d_{\\mathrm{chunk}}=2$ we have an average running time $0.0127$ seconds and with $d_{\\mathrm{chunk}}=4$ we have an average running time $0.6343$ seconds;
for $\\varepsilon = 10$, with $d_{\\mathrm{chunk}}=2$ the average running time are $0.0128$ seconds and with $d_{\\mathrm{chunk}}=4$ the average running time are $0.7301$ seconds.
More data on average running time against $\\varepsilon\\in [0.06, 10]$ with $d_{\\mathrm{chunk}}=4$ are plotted in Figure A in the submitted pdf file.

In conclusion, we believe the reviews for our contributions are overall positive.
We sincerely hope that we have adequately addressed the reviewers' concerns and questions.
We have also submitted another file in pdf format to share additional figures.
Any further feedback and discussion is gladly welcomed.
If our rebuttal effectively addresses the reviewers' concerns, we kindly invite them to consider updating their scores. Thank you very much for your time and feedback.

**References**

[Erlingsson, Úlfar, et al.] "Amplification by shuffling: From local to central differential privacy via anonymity," SODA 2019.

---

### Decision · Program_Chairs · 2024-09-25

**Decision:**

Accept (poster)

**Comment:**

This submission provides a new method for compressing local differential private reports. It is based on Poisson functional representations, a tool from information theory that allows encoding a random variable in close to the information theoretic minimum expected number of bits in a universal manner that does not require the decoder to know the probability distribution of the random variable.
The paper presents a generalization called Poisson Private Representations (PPR) that preserves the differential privacy guarantee up to a multiplicative factor, and has similar encoding efficiency. Unlike previous local differential privacy compression techniques, PPR result in exactly the same distribution of decoded values as the original uncompressed local differentially private protocol, so it preserves properties such as unbiasedness. The paper investigates the utility of PPR in a couple of settings including metric privacy and distributed mean estimation.

There is agreement among the reviewers that the paper should be accepted. The authors are expected to implement in the final version the changes that were promised in the rebuttal.

Moreover, in the AC-reviewers discussion, the following point came up:

The authors seem to be missing part of the literature on shuffle DP. Specifically, the rebuttal said:

"without exact simulation we can only rely on generic privacy amplification techniques, such as shuffling [Erlingsson et al., 2019] (known to suffer from highly sub-optimal constants and only meaningful for limited privacy regimes where epsilon_local << 1), giving a much looser central DP."

This is incorrect the literature following up on Erlingsson et al., 2019 managed to get much tighter bounds that work for epsilon_local >> 1. In particular, please see the paper of Feldman et al. (FOCS 2021): https://vtaly.net/papers/FMT-Clones-0921.pdf

The authors should clarify this in the final version.